# Unnatural Error Correction: GPT-4 Can Almost Perfectly Handle Unnatural Scrambled Text

**Qi Cao, Takeshi Kojima, Yutaka Matsuo** and **Yusuke Iwasawa**

The University of Tokyo, Japan

`{qi.cao,t.kojima,matsuo,iwasawa}@weblab.t.u-tokyo.ac.jp`

## Abstract

While Large Language Models (LLMs) have achieved remarkable performance in many tasks, much about their inner workings remains unclear. In this study, we present novel experimental insights into the resilience of LLMs, particularly GPT-4, when subjected to extensive character-level permutations. To investigate this, we first propose the *Scrambled Bench*, a suite designed to measure the capacity of LLMs to handle scrambled input, in terms of both recovering scrambled sentences and answering questions given scrambled context. The experimental results indicate that multiple advanced LLMs demonstrate the capability akin to *typoglycemia* [1], a phenomenon where humans can understand the meaning of words even when the letters within those words are scrambled, as long as the first and last letters remain in place. More surprisingly, we found that only GPT-4 nearly flawlessly processes inputs with *unnatural errors*, a task that poses significant challenges for other LLMs and often even for humans. Specifically, GPT-4 can almost perfectly reconstruct the original sentences from scrambled ones, decreasing the edit distance by 95%, even when all letters within each word are entirely scrambled. It is counter-intuitive that LLMs can exhibit such resilience despite severe disruption to input tokenization caused by scrambled text. [2]

## 1 Introduction

Large language models (LLMs) demonstrate impressive proficiency across a range of tasks, with certain capabilities emerging as the models scale up in size—a phenomenon commonly known as *emergent abilities*. (Wei et al., 2022a). As the LLMs

---

[1]Typoglycemia is an internet meme without specific scientific backing. Nonetheless, people do exhibit the ability to recognize scrambled common words to some extent. Our focus is not on examining this ability of humans, we use the concept for illustrative purposes.

[2]Code will be available at https://github.com/ccqq77/unnatural-error-correction.

Figure 1: GPT-4 can recover the original sentence from the scrambled sentence, even if the tokenization drastically changes. (The colors indicate the division of sub-words during the tokenization.)

become more "intelligent", many new benchmarks have been proposed (Liang et al., 2022; Qin et al., 2023) to investigate the ability of LLMs. Nonetheless, the internal mechanisms underlying the capabilities of LLMs remain enigmatic.

Several studies investigate the behavior of LLMs given some input perturbations. For example, Sinha et al. (2021a,b); Abdou et al. (2022) investigate the influence of *word-level* permutations and show that models are insensitive to permutations of word order that corrupt the original syntax, in some downstream tasks (e.g., natural language inference). These results are particularly interesting because they challenge the common assumption of the inner workings of LLMs, i.e., LLMs understand human-like syntax to some extent and use it to understand sentences.

Going beyond the effect of word-level permutations, we investigate the ability of LLMs under *character-level* permutations. LLMs are supposed to rely on the tokenizers to turn natural language into the form that LLMs can perceive. It would be counter-intuitive if LLMs could effectively handle text containing *unnatural* permutations that significantly alter tokenization. In other words, we propose the following research question: *Deos the*

*oredr of ltteers in wrods mttaer for LLMs?* Note that the above sentence contains scrambled words, but humans can somehow recognize and understand such a sentence, as several cognitive studies have explored (Rawlinson, 2007; Mason, 1982; Johnson and Eisler, 2012). Analyzing the robustness of LLMs against such character-level permutations can shed light on their word comprehension capabilities and reveal differences between various LLMs and human understanding.

To this end, this paper first constructs *Scrambled Bench*, which converts existing benchmarks into a test suite to measure the ability of LLMs to handle scrambled text. We designed two types of tasks: (1) *Scrambled Sentence Recovery*, which tests the capability of LLMs to reconstruct the original sentences from scrambled ones, and (2) *Scrambled Question Answering*, which measures how well LLMs can answer questions when some context is scrambled. Note that since the slight change in letter-order within a word drastically changes the tokenized output (see Figure 1) , it is questionable whether LLMs can recognize the scrambled words in a sentence. Counter-intuitively, we show that the most powerful LLMs are able to handle scrambled sentences to varying degrees, when we scramble words while keeping the first and last letters unchanged. More surprisingly, we found that only GPT-4 can almost flawlessly process inputs with *unnatural errors*, even under extreme conditions. That is, even when we scramble all letters in words, GPT-4 manages to handle such input — a significantly challenging task for other models and even humans. For instance, GPT-4 can reconstruct the original sentences to near-perfect recovery rate in the extreme scenario, as in Figure 1.

## 2   Related Works

The most related works are the studies investigating the effects of word or sub-word level perturbations (Sinha et al., 2021a,b; Pham et al., 2021; Abdou et al., 2022) and the studies evaluating the robustness of LLMs (Wang et al., 2023; Zhu et al., 2023). To the best of our knowledge, no existing studies have investigated LLMs' ability to handle character-level permutations, particularly those of an extremely high level that drastically change tokenization. Our study aims to fill this gap. Table 2 in Appendix B categorizes the prior studies and demonstrates the position of our study. Besides, we directly evaluate the ability to recover the scram-

bled text along with the task accomplishment given scrambled context. It differs with typographical error correction (Shah and de Melo, 2020; Sun et al., 2022), as (i) we do not train models to correct errors, i.e., we measure the ability of LLMs, and (ii) we add much more severe noises than natural typographical errors. The word unscrambling task in BigBench (Srivastava et al., 2023) is similar to our recovery task. However, it is more akin to a wordplay puzzle rather than a task for comprehending scrambled text, since it includes only single common words and no context is given.

## 3   Scrambled Bench

We propose two tasks to evaluate the capability of LLMs to handle scrambled text.

**(1) Scrambled Sentence Recovery (ScrRec).**   In this task, we provide a sentence containing scrambled words to LLMs and then ask them to recover the original sentence from it. This task can be utilized to directly measure the capability of LLMs to recognize and reconstruct the scrambled words in a sentence.

**(2) Scrambled Question Answering (ScrQA).** While ScrRec can directly measure the capability to comprehend and process scrambled text, it is an "unusual" task for LLMs. If a model does not perform well on the recovery task, there are two possible reasons: (i) having difficulty following the instructions and (ii) not being able to recover sentences. To distinguish them, we measure the ability to accomplish a standard task (i.e., QA) given scrambled context. Specifically, we scramble the content that contains essential information for answering questions and then assess the models based on the variations in their performances.

### 3.1   Dataset Creation

In this study, we primarily utilize a scrambled version of RealtimeQA (Kasai et al., 2022) for evaluation. A common issue in evaluating LLMs is data contamination, which occurs when the test data of downstream tasks is present in the training data. It particularly affects our experiments because the analysis would be useless if some models memorized the original contents. RealtimeQA is a dynamic question answering dataset that weekly announces questions about recent news that are unlikely to be memorized by the current LLMs. Specifically, we collect the most recent

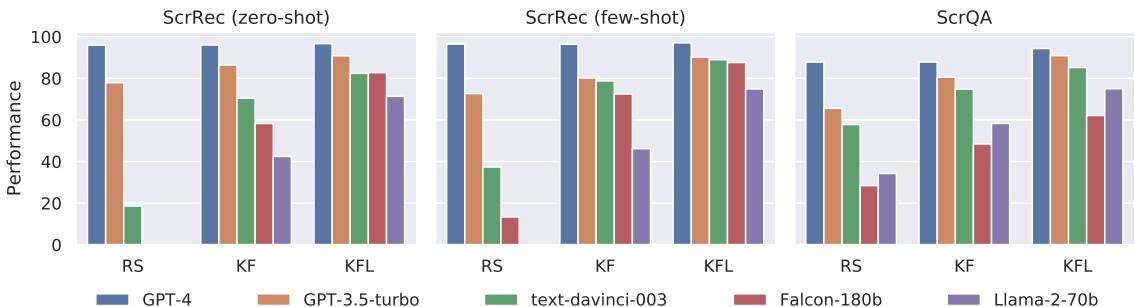

Figure 2: Experimental results on scrambled RealtimeQA dataset. Performance on ScrRec with a zero-shot setting (left). Metric: RR(%). Performance on ScrRec with a few-shot setting (middle). Metric: RR(%). Performance on ScrQA with a zero-shot setting (right). Metric: RPG(%). The missing bars represent values less than zero.

data (2023/03/17–2023/08/04) from RealtimeQA (totally 419 samples) and process the evidence sentences to construct samples for ScrRec and ScrQA. Finally, 418 samples are selected for ScrRec (removing a duplicate sentence), and 346 samples are selected for ScrQA (manually eliminating 73 samples when the provided evidence does not provide sufficient information to answer the corresponding question).

Besides, we also introduce two additional datasets: DREAM (Sun et al., 2019) and AQuA-RAT (Ling et al., 2017). DREAM is a dialogue-based multiple-choice reading comprehension dataset. AQuA-RAT is a dataset of math word problems necessitating multi-step reasoning for their resolution. For DREAM dataset, we constructed the dataset by selecting 1025 samples with annotated categories from the development and test sets and then scrambling the dialogue part of each question. For AQuA-RAT dataset, we adopt the few-shot Chain of Thought (CoT) setting as in Wei et al. 2022b and evaluate LLMs with scrambled questions in samples and demonstrations.

For each dataset, we generate scrambled text with various scramble types and rates. **(1) Randomly Scramble (RS)**. For each sentence, we randomly select a certain percentage (20%, 50%, 100% in our case[3]) of words and randomly shuffle the positions of letters in each selected word (Arabic numerals are kept invariant). **(2) Keep First (KF)**. We keep the first letter in each word unchanged and randomly shuffle the letters in other positions. **(3) Keep First and Last (KFL)**. We keep the first and last letter in each word unchanged and randomly shuffle the letters in other positions. We use KF and KFL to investigate the effects of

---

[3]In the latter part of the paper, RS denotes 100% randomly scrambling, unless a different percentage is specified.

different scramble types in word recognition for LLMs.

### 3.2 Metrics

The average Edit Distance (ED) (Levenshtein, 1966) between the original sentences and the recovered sentences is a natural metric to quantify the performance on ScrRec. Besides, we define Recovery Rate (RR) to measure the proportion of ED reduced in recovered sentences, which makes the performance comparison on different settings more straightforward as:

$$RR = \frac{\sum_i^n ED(ori_i, rec_i) - \sum_i^n ED(ori_i, scr_i)}{\sum_i^n ED(ori_i, scr_i)} \tag{1}$$

where $ori_i$, $scr_i$, $rec_i$ mean the original sentence, the scrambled sentence, and the recovered sentence of sample $i$, respectively. If a model is capable of recovering scrambled text to some extent, its RR should range between 0 and 100%.

For ScrQA, accuracy is a natural metric to measure performance. But varying capabilities of models on original questions make it hard to compare the performance among models. So, Relative Performance Gain (RPG) is defined to mitigate the differences and make evaluations focus on how well models can extract information from scrambled text in comparison to original text as:

$$RPG = \frac{Acc_{scr} - Acc_{sub}}{Acc_{ori} - Acc_{sub}} \tag{2}$$

where $Acc_{ori}$, $Acc_{scr}$, $Acc_{sub}$ are defined as accuracy when using original evidences, scrambled evidences, substituted evidences (in which all letters of words that can be scrambled have been randomly substituted), respectively. $Acc_{ori}$, $Acc_{sub}$ are served as the higher bound and the lower bound of accuracy, respectively. Normally, RPG ranges between 0 and 100%.

# 4 Experiments

**Models and Settings.** In the experiments, we evaluate the most powerful closed-source LLMs, including text-davinci-003 (Brown et al., 2020), GPT-3.5-turbo and GPT-4 (OpenAI, 2023) and the open-source models from Falcon series (Penedo et al., 2023), Llama-2 series (Touvron et al., 2023), MPT series (Team, 2023), UL2 series (Tay et al., 2022), T5 series (Raffel et al., 2020; Chung et al., 2022; Xue et al., 2022). In scrambled RealtimeQA dataset, we adopt a zero-shot setting and a few-shot setting with 3-shot exemplars from the wikiQA dataset (Yang et al., 2015) for ScrRec, while we only conduct experiments on a zero-shot setting (since the task is rather straightforward) for ScrQA. In scrambled DREAM dataset, the setting is also zero-shot ScrQA. In scrambled AQuA dataset, we adopt a few-shot CoT setting with scrambled demonstrations (in the question part). Note that we are showcasing the results of the top five most proficient LLMs (i.e., GPT-4, GPT-3.5-turbo, text-davinci-003, Falcon-180b, Llama-2-70b) in this section, but comprehensive results can be found in Appendix C.

**Results 1: Effect of different scramble types.** Figure 2 show the results on zero-shot ScrRec, few-shot ScrRec, and ScrQA, with three scramble types: randomly scramble (RS), keep first (KF), and keep first and last (KFL). The results show the performance gaps among models are not large in KFL setup. However, except for GPT-4, performance significantly decreases as the difficulty of scramble types increases (KFL, KF, and RS in order). In contrast, the performance of GPT-4 remains constantly high regardless of the scramble types. For ScrRec, RR of GPT-4 is constantly above 95% for all setups. For ScrQA, GPT-4 also constantly performs best with very limited accuracy drop, as the difficulty of scramble types increases.

**Results 2: Effect of different scramble rates.** Figure 3 illustrates the relationship between the scramble rates (i.e., the percentages of randomly scrambled words in text) and the performance on ScrRec with scrambled RealtimeQA. As the scramble rates increases, RR decreases for text-davinci-003, Falcon-180b and Llama-2-70b. RR of GPT-3.5-turbo and GPT-4 does not change significantly. GPT-4 outperforms other models by a wide margin, with higher than 95% RR for most setups (except for 20% scramble rate).

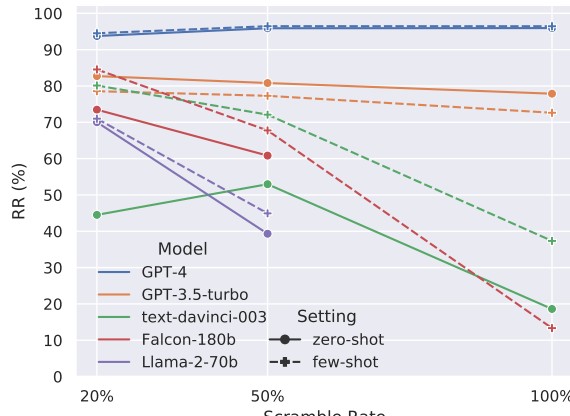

Figure 3: The effect of scramble rates on the performance on ScrRec-RealtimeQA. The missing points represent values less than zero.

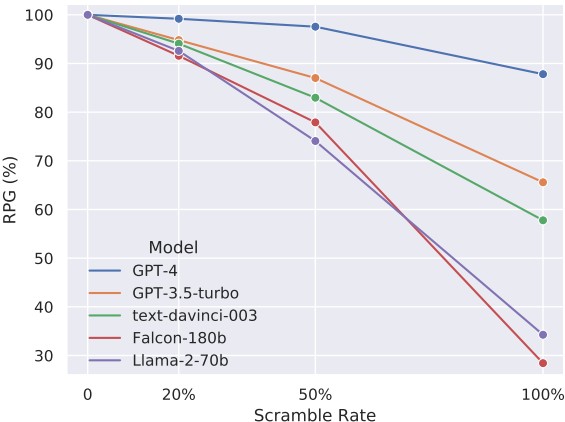

Figure 4: The effect of scramble rates on the performance on ScrQA-RealtimeQA.

Similarly, Figure 4 plots RPG against the scramble rates for different models on ScrQA with scrambled RealtimeQA. RPG of all models drops as scramble percentage increases. But GPT-4 maintains 87.8% of its original performance even with 100% scrambled evidence. The gap between the performance of different models becomes larger when the scramble rates increases. Table 3, Table 4 and Table 5 in Appendix C show full results on scrambled RealtimeQA dataset, with additional evaluation metrics (ED for ScrRec and Acc for ScrQA).

**Results 3: Results of additional datasets.** Finally, we test the generality of the finding across datasets by two additional datasets for ScrQA. For scrambled DREAM dataset, we evaluate performance not only overall but also on different categories of questions, using the annotations. The performance disparities between GPT-4 and other models are more pronounced than those observed on RealtimeQA, possibly since DREAM requires

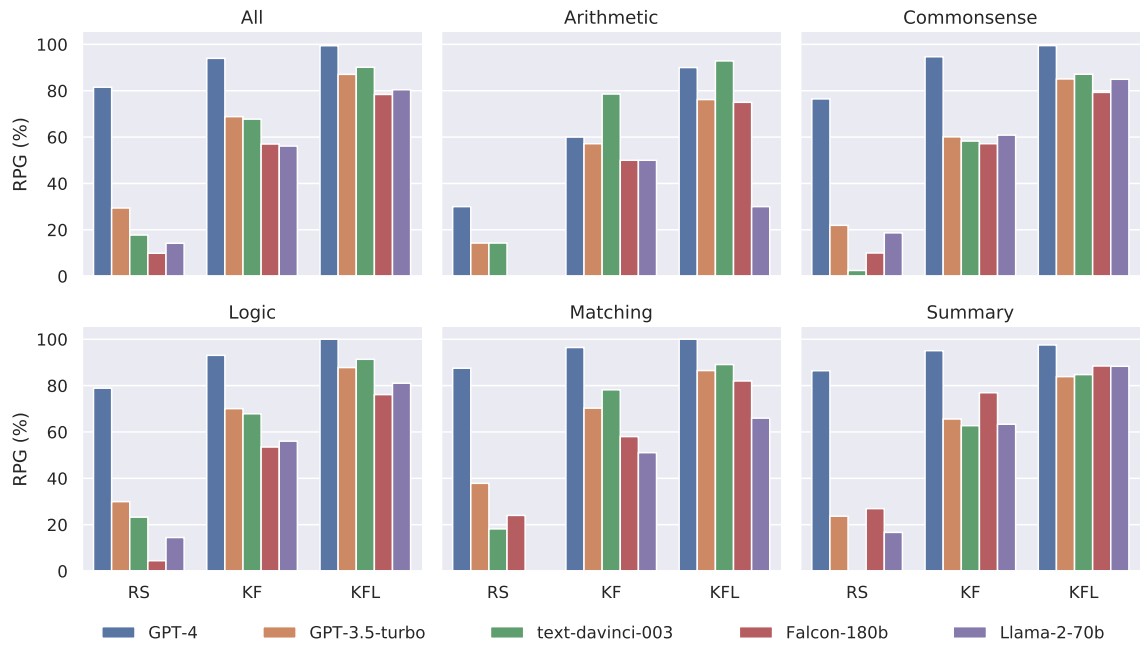

Figure 5: Experimental results of ScrQA on scrambled DREAM dataset. Overall performance and performance on different categories of questions. The missing bars represent values less than or equal to zero.

| | Exemplars | Original | 20% | 50% | 100% |
|---|---|---|---|---|---|
| **Question** | | | | | |
| **Original** | | 46.85 ǀ 52.76 ǀ 66.93 | 47.24 ǀ 51.97 ǀ 70.87 | 45.67 ǀ 54.72 ǀ 72.05 | 46.46 ǀ 54.72 ǀ 70.08 |
| **20%** | | 40.94 ǀ 49.61 ǀ 66.54 | 40.16 ǀ 51.57 ǀ 74.02 | 42.52 ǀ 52.36 ǀ 68.50 | 45.28 ǀ 50.79 ǀ 71.26 |
| **50%** | | 40.94 ǀ 40.94 ǀ 67.72 | 41.34 ǀ 44.88 ǀ 65.75 | 40.16 ǀ 47.64 ǀ 64.96 | 44.09 ǀ 42.91 ǀ 67.32 |
| **100%** | | 26.77 ǀ 29.92 ǀ 64.17 | 27.17 ǀ 31.89 ǀ 62.60 | 33.47 ǀ 27.17 ǀ 64.96 | 28.35 ǀ 35.04 ǀ 60.24 |

Table 1: Experimental results of ScrQA on scrambled AQuA-RAT dataset with a few-shot CoT setting. LLMs are assessed in various scenarios by adjusting scramble rate of questions in examplars and the main question. (left: text-davinci-003; middle: GPT-3.5-turbo; right: GPT-4). Metric: Acc (%).

higher-level comprehension of longer texts. Performance on arithmetic questions tends to be more susceptible to scrambled text compared to other categories, even for GPT-4. Table 1 demonstrates experimental results with a 4-shot CoT setting on scrambled AQuA-RAT dataset (we only test the performance of three closed-source models here because even the original questions in AQuA-RAT are too challenging for most open-source models). We evaluate models with several different scenarios by changing scramble rates of both questions in examplars and main question. The results show the influence of scrambled examplars is relatively small. And the performance of GPT-3.5-turbo and text-davinci-003 drop significantly when the scramble rate of main questions increases to 100%, while GPT-4 remains most of its original performance. Table 6, Table 7 and Table 8 in Appendix C show full results on scrambled DREAM dataset.

## 5 Conclusion

In this study, we propose *Scrambled Bench*, a test suite to measure the ability of LLMs to handle scrambled text, including two tasks (i.e., scrambled sentence recovery and scrambled question answering) and construct scrambled datasets based on RealtimeQA, DREAM and AQuA-RAT. Despite the scrambled text drastically changes the tokenization, we demonstrate that advanced LLMs are capable of processing scrambled text to varying degrees. However, most LLMs have difficulty handling text that is scrambled to an extreme degree (i.e., 100% randomly scrambling). Surprisingly, for both tasks, GPT-4 shows good results and outperforms other models by a large margin. For the scrambled sentence recovery task, GPT-4 can recover sentences by 95% edit distance reduction even in 100% randomly scrambling settings. For the scrambled question answering task, GPT-4 can maintain a very high proportion of its original accuracy using scrambled context.

## Limitations

For LLMs, there are various ways to disrupt the tokenization of words (e.g., inserting letters, substituting letters). In this study, we only investigate the influence of scrambling the letter-order in words. Investigating the performance of LLMs to handle other situations would be an interesting topic.

In addition, we have conducted our experiments using only three datasets, RealtimeQA, DREAM and AQuA-RAT. Experiments on more diverse datasets could be another future work. Note that the two tasks can be applicable for diverse datasets, and it is easy to extend the analysis.

We investigate the capability of different LLMs to handle scrambled text in different settings. However, it is difficult to conclude the reason why (some) LLMs are capable to these tasks. Especially, the reason why GPT-4 can perform almost perfectly would be an interesting topic worth further investigation. We can not access the closed-source models directly and are aware of little information about them (even the exact model size of GPT-4). These situation make investigating the reason difficult. An hypothesis is that this capability might be related to training methods, such as incorporating tasks similar to denoising in the training objectives, or using a vast amount of text data containing various errors in the training process. Another hypothesis is that this capability emerges as LLMs scale. Validating these hypotheses could potentially enhance our understanding of the inner workings of LLMs, thereby enabling us to reverse-engineer and recreate more sophisticated models like GPT-4.

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

# A Examples of different tasks (by GPT-4)

## A.1 Zero-shot ScrRec on RealtimeQA

**Input:**
The following sentence contains words with scrambled letters. Please recover the original sentence from it.
Scrambled sentence: A reGebr byba ulfaorm wsa titbudiserd ot soetsr epdstie a lclera eovr bslpioes ionmanantitco, grnoadicc ot eth ADF. heT pyomacn si noniacrggue rptsean ot ckhec yna poducsrt yhte evah ta mhoe nda cdisadr sehot taht aym eb ecaeftdf.
Recovered sentence:
**Output:**
A Gerber baby formula was distributed to stores despite a recall over possible contamination, according to the FDA. The company is encouraging parents to check any products they have at home and discard those that may be affected.

## A.2 Few-shot ScrRec on RealtimeQA

**Input:**
Scrambled Sentence: eTh camp continued to fnctinuo this ayw ilntu the rwa needd.
Recovered Sentence: The camp continued to function this way until the war ended.

Scrambled Sentence: It swa first developed ni the 1980s yb oAcrn Computers tdL ot erowp their pstodke nmecisah and subsequently supn off sa a separate paocnmy, now ARM Holdings.
Recovered Sentence: It was first developed in the 1980s by Acorn Computers Ltd to power their desktop machines and subsequently spun off as a separate company, now ARM Holdings.

Scrambled Sentence: According to the CIA kcbFotoa, the United States is one fo eethr iusecnort (het etrhos nebgi Liberia nda mBuar/Myanmar) that sha not adopted eth International System fo Utins (SI) rmtcei symset as iethr ffliicao system fo gswheit dna measures.
Recovered Sentence: According to the CIA Factbook, the United States is one of three countries (the others being Liberia and Burma/Myanmar) that has not adopted the International System of Units (SI) metric system as their official system of weights and measures.

Scrambled sentence: A reGebr byba ulfaorm wsa titbudiserd ot soetsr epdstie a lclera eovr

bslpioes ionmanantitco, grnoadicc ot eth ADF. heT pyomacn si noniacrggue rptsean ot ckhec yna poducsrt yhte evah ta mhoe nda cdisadr sehot taht aym eb ecaeftdf.
Recovered sentence:
**Output:**
A Gerber baby formula was distributed to stores despite a recall over possible contamination, according to the FDA. The company is encouraging parents to check any products they have at home and discard those that may be affected.

## A.3 Zero-shot ScrQA on RealtimeQA

**Input:**
Question: Which type of product was recently distributed to some US stores despite a recall notice?
Choices: (A)Salad dressing (B)Baby formula (C)Ground beef (D)Whole milk
Evidence: A reGebr byba ulfaorm wsa titbudiserd ot soetsr epdstie a lclera eovr bslpioes ionmanantitco, grnoadicc ot eth ADF. heT pyomacn si noniacrggue rptsean ot ckhec yna poducsrt yhte evah ta mhoe nda cdisadr sehot taht aym eb ecaeftdf.
Answer: Based on the evidence, among A through D, the answer is
**Output:**
(B)Baby formula

## A.4 Zero-shot ScrQA on DREAM

**Input:**
Dialogue:
M: owH goln veah uyo eben ehctnaig ni iths edmdli colosh?
W: Fro ent esray. oT eb akfrn, I'm edirt fo higneact het mesa bokoextt rof os ongl hghuto I od jyneo gebin a aterceh. I'm idenoncisrg gntriy minseghot wne.
Question: What's the woman probably going to do?
Choices: (A)To teach a different textbook. (B)To change her job. (C)To learn a different textbook.
Answer: Based on the dialogue, among A through C, the answer is
**Output:**
(B)To change her job.

## A.5 Few-shot CoT ScrQA on AQuA-RAT

**(1)**
**Input:**
Question: hJno nfodu ttah teh gaeerav fo 15 rnsbmue si 40. fI 10 si edadd ot ceha umenbr enht

| Level | Natural errors | Example (Original: How does the change affect tokenization?) | Change of tokenization | Severity | Studies |
|---|---|---|---|---|---|
| Word | No | How the affect does tokenization change? | Order | Low–High | Sinha et al. 2021a, Sinha et al. 2021b, Pham et al. 2021, Abdou et al. 2022 |
| Sub-word | No | How change token does affectization the? | Order | Low–High | Abdou et al. 2022 |
| Character | Yes | How does the chagne affect tokeniztion? | Tokens | Low | Wang et al. 2023, Zhu et al. 2023 |
| Character | No | wHo deso teh chenga afctfe aotznoenitki? | Tokens | Middle–Extremely High | Ours |

Table 2: Categorizing the prior works and demonstrating the position of this work.

teh naem fo het rbenmsu si?
Choices: (A)50 (B)45 (C)65 (D)78 (E)64
Answer: If 10 is added to each number, then the mean of the numbers also increases by 10. So the new mean would be 50. The answer is (A).

Question: fI a / b = 3/4 dan 8a + 5b = 22, htne nfid het evual fo a.
Choices: (A)1/2 (B)3/2 (C)5/2 (D)4/2 (E)7/2
Answer: If a / b = 3/4, then b = 4a / 3. So 8a + 5(4a / 3) = 22. This simplifies to 8a + 20a / 3 = 22, which means 44a / 3 = 22. So a is equal to 3/2. The answer is (B).

Question: A nprseo si gtlanrvie ta 20 mk/rh nda hdrecae ihs ynedits ni 2.5 rh neth fdin eth dicatsen?
Choices: (A)53 km (B)55 km (C)52 km (D)60 km (E)50 km
Answer: The distance that the person traveled would have been 20 km/hr * 2.5 hrs = 50 km. The answer is (E).

Question: wHo aymn etosreyksk ear deeend ot peyt het nsrbmeu rfmo 1 ot 500?
Choices: (A)1156 (B)1392 (C)1480 (D)1562 (E)1788
Answer: There are 9 one-digit numbers from 1 to 9. There are 90 two-digit numbers from 10 to 99. There are 401 three-digit numbers from 100 to 500. 9 + 90(2) + 401(3) = 1392. The answer is (B).

Question: A ritna inunrgn ta a eepsd fo 100 mlesi/uhro, eakts 10 hurso ot cerah ist dneitiotsan. trefA ovgeirnc etarqru fo het cntsdaie, ti tstrsa grniina nda eth rniat ash ot eb dlseow ot seedp fo

75 liesm/uhor. Wtha si het otalt orjeyun iroatdnu?
Choices: (A)10 (B)11.5 (C)12.5 (D)13.5 (E)15
Answer:
**Output:**
The train covers a quarter of the distance at 100 miles/hour, so it travels for 2.5 hours (10 hours / 4) at this speed. The remaining distance is 3/4 of the total distance, which takes the train 7.5 hours (10 hours - 2.5 hours) at 100 miles/hour. When the train slows down to 75 miles/hour, it takes 1.5 times longer to cover the same distance (100 miles/hour / 75 miles/hour = 1.33). So the train takes 11.25 hours (7.5 hours * 1.33) to cover the remaining distance at 75 miles/hour. The total journey duration is 13.75 hours (2.5 hours + 11.25 hours). The answer is (D).

**(2)**
**Input:**
Question: John found that the average of 15 numbers is 40. If 10 is added to each number then the mean of the numbers is?
Choices: (A)50 (B)45 (C)65 (D)78 (E)64
Answer: If 10 is added to each number, then the mean of the numbers also increases by 10. So the new mean would be 50. The answer is (A).

Question: If a / b = 3/4 and 8a + 5b = 22, then find the value of a.
Choices: (A)1/2 (B)3/2 (C)5/2 (D)4/2 (E)7/2
Answer: If a / b = 3/4, then b = 4a / 3. So 8a + 5(4a / 3) = 22. This simplifies to 8a + 20a / 3 = 22, which means 44a / 3 = 22. So a is equal to 3/2. The answer is (B).

Question: A person is traveling at 20 km/hr

and reached his destiny in 2.5 hr then find the distance?
Choices: (A)53 km (B)55 km (C)52 km (D)60 km (E)50 km
Answer: The distance that the person traveled would have been 20 km/hr * 2.5 hrs = 50 km. The answer is (E).

Question: How many keystrokes are needed to type the numbers from 1 to 500?
Choices: (A)1156 (B)1392 (C)1480 (D)1562 (E)1788
Answer: There are 9 one-digit numbers from 1 to 9. There are 90 two-digit numbers from 10 to 99. There are 401 three-digit numbers from 100 to 500. 9 + 90(2) + 401(3) = 1392. The answer is (B).

Question: ieahMnc A psut otu a oy-oy ryeve 6 nsimtue. enicMah B pstu uto a oy-oy veeyr 9 nmusite. tAefr hwo aynm smtnuei lwil yeth hvea eurdpcod 10 oy-soy?
Choices: (A)24 minutes (B)32 minutes (C)36 minutes (D)64 minutes (E)72 minutes
Answer:
**Output:**
First, let's rewrite the question with the words unscrambled: "Machine A puts out a yo-yo every 6 minutes. Machine B puts out a yo-yo every 9 minutes. After how many minutes will they have produced 10 yo-yos?"

To solve this problem, we can find the least common multiple (LCM) of 6 and 9, which is 18. This means that every 18 minutes, both machines will have produced a yo-yo at the same time. In 18 minutes, Machine A will have produced 3 yo-yos (18/6) and Machine B will have produced 2 yo-yos (18/9), for a total of 5 yo-yos. Since we need 10 yo-yos, we can multiply 18 minutes by 2 to get 36 minutes. The answer is (C) 36 minutes.

## B  Summary of related work

Table 2 categorizes the related work and demonstrates the position of our study.

## C  Full experimental results

We conduct experiments using the most powerful closed-source LLMs, including text-davinci-003, GPT-3.5-turbo and GPT-4 and various open-source LLMs, including the models from Falcon series, Llama-2 series, MPT series, UL2 series, and T5 series. The open-source model covers diverse model architectures (decoder only and encoder-decoder), model size (from 7b to 180b), training objectives (e.g., with or without further finetuning) and tokenizers (e.g., tokenizer-free: ByT5-xxl).

For GPT-4, the version GPT-4-0314 is used. For GPT-3.5-turbo, the version GPT-3.5-turbo-0301 is used. For Falcon-180b and Falcon-180b-chat, the quantized method (Dettmers et al., 2023) is used to load the model and run the experiments. It probably affects their performance to some extent.

### C.1  Full experimental results on scrambled RealtimeQA dataset

Table 3 and Table 4 illustrates the full experimental results on scrambled RealtimeQA dataset for few-shot ScrRec and zero-shot ScrRec, respectively.

Table 5 illustrates the full experimental results on scrambled RealtimeQA dataset for zero-shot ScrQA.

### C.2  Full experimental results on scrambled DREAM dataset

Table 6 illustrates the full experimental results on scrambled DREAM dataset for zero-shot ScrQA.

Table 7 and Table 8 illustrates the experimental results on different question types of the top five models (like in Figure 5) with accuracy and RPG as the metrics, respectively.

| Model | 20% | 50% | 100% | KF | KFL |
|---|---|---|---|---|---|
| GPT-4 | 1.49 | 2.40 | 4.79 | 3.79 | 2.24 |
| | 94.51 | 96.46 | 96.45 | 96.37 | 97.04 |
| GPT-3.5-turbo | 5.80 | 15.38 | 36.87 | 20.76 | 7.40 |
| | 78.57 | 77.29 | 72.63 | 80.14 | 90.22 |
| text-davinci-003 | 5.38 | 18.89 | 84.39 | 22.21 | 8.36 |
| | 80.13 | 72.11 | 37.35 | 78.75 | 88.95 |
| Falcon-180b | 4.18 | 21.84 | 116.73 | 28.77 | 9.40 |
| | 84.55 | 67.76 | 13.34 | 72.47 | 87.58 |
| Falcon-180b-chat | 11.61 | 38.19 | 186.65 | 63.51 | 21.73 |
| | 57.12 | 43.62 | < 0 | 39.24 | 71.28 |
| Falcon-40b | 17.27 | 69.86 | 185.26 | 119.00 | 66.03 |
| | 36.18 | < 0 | < 0 | < 0 | 12.71 |
| Falcon-40b-instruct | 18.79 | 74.26 | 192.86 | 130.82 | 62.39 |
| | 30.56 | < 0 | < 0 | < 0 | 17.53 |
| Llama-2-70b | 7.85 | 37.30 | 169.85 | 56.25 | 18.99 |
| | 71.0 | 44.93 | < 0 | 46.19 | 74.9 |
| Llama-2-70b-chat | 11.11 | 54.49 | 196.44 | 106.31 | 43.18 |
| | 58.97 | 19.56 | < 0 | < 0 | 42.93 |
| Llama-2-13b | 14.18 | 59.13 | 192.67 | 103.50 | 42.67 |
| | 47.62 | 12.71 | < 0 | 0.99 | 43.59 |
| Llama-2-13b-chat | 20.75 | 73.04 | 201.50 | 119.52 | 56.03 |
| | 23.35 | < 0 | < 0 | < 0 | 25.93 |
| Llama-2-7b | 20.90 | 79.83 | 194.48 | 135.50 | 68.18 |
| | 22.77 | < 0 | < 0 | < 0 | 9.88 |
| Llama-2-7b-chat | 25.14 | 83.13 | 196.88 | 132.44 | 73.84 |
| | 7.11 | < 0 | < 0 | < 0 | 2.39 |
| MPT-30b | 14.36 | 54.81 | 181.43 | 90.92 | 36.14 |
| | 46.96 | 19.08 | < 0 | 13.02 | 52.23 |
| MPT-30b-instruct | 19.36 | 66.05 | 170.88 | 99.38 | 50.44 |
| | 28.46 | 2.49 | < 0 | 4.92 | 33.33 |
| UL2 | 18.12 | 68.79 | 153.53 | 109.47 | 71.70 |
| | 33.03 | < 0 | < 0 | < 0 | 5.22 |
| Flan-UL2 | 20.51 | 62.91 | 137.15 | 105.67 | 74.35 |
| | 24.21 | 7.13 | < 0 | < 0 | 1.72 |
| Flan-T5-xxl | 16.60 | 59.34 | 136.41 | 105.44 | 73.97 |
| | 38.68 | 12.4 | < 0 | < 0 | 2.22 |
| ByT5-xxl | 79.27 | 104.28 | 160.17 | 123.97 | 102.50 |
| | < 0 | < 0 | < 0 | < 0 | < 0 |

Table 3: The experimental results on the few-shot ScrRec using RealtimeQA dataset (upper: ED; lower: RR).

| Model | 20% | 50% | 100% | KF | KFL |
|---|---|---|---|---|---|
| GPT-4 | 1.69 | 2.78 | 5.47 | 4.19 | 2.53 |
| | 93.76 | 95.89 | 95.94 | 95.99 | 96.66 |
| GPT-3.5-turbo | 4.68 | 13.00 | 29.80 | 14.22 | 6.94 |
| | 82.72 | 80.82 | 77.88 | 86.39 | 90.82 |
| text-davinci-003 | 15.02 | 31.86 | 109.63 | 30.88 | 13.30 |
| | 44.52 | 52.96 | 18.61 | 70.45 | 82.41 |
| Falcon-180b | 7.18 | 26.53 | 159.36 | 43.63 | 13.08 |
| | 73.48 | 60.84 | < 0 | 58.25 | 82.72 |
| Falcon-180b-chat | 15.92 | 50.50 | 180.13 | 76.31 | 29.29 |
| | 41.17 | 25.46 | < 0 | 27.0 | 61.29 |
| Falcon-40b | 20.59 | 66.33 | 177.33 | 92.45 | 39.04 |
| | 23.94 | 2.08 | < 0 | 11.56 | 48.39 |
| Falcon-40b-instruct | 26.42 | 78.94 | 178.55 | 104.46 | 48.42 |
| | 2.4 | < 0 | < 0 | 0.07 | 36.0 |
| Llama-2-70b | 8.11 | 41.10 | 180.82 | 60.16 | 21.68 |
| | 70.04 | 39.32 | < 0 | 42.45 | 71.34 |
| Llama-2-70b-chat | 12.65 | 60.23 | 181.30 | 118.13 | 51.73 |
| | 53.26 | 11.08 | < 0 | < 0 | 31.62 |
| Llama-2-13b | 20.21 | 75.54 | 231.41 | 127.63 | 59.77 |
| | 25.32 | < 0 | < 0 | < 0 | 20.99 |
| Llama-2-13b-chat | 26.79 | 84.09 | 191.85 | 120.51 | 63.82 |
| | 1.03 | < 0 | < 0 | < 0 | 15.64 |
| Llama-2-7b | 29.81 | 106.38 | 192.05 | 165.56 | 88.65 |
| | < 0 | < 0 | < 0 | < 0 | < 0 |
| Llama-2-7b-chat | 48.92 | 123.66 | 202.16 | 167.49 | 112.03 |
| | < 0 | < 0 | < 0 | < 0 | < 0 |
| MPT-30b | 20.54 | 70.69 | 226.34 | 135.58 | 62.16 |
| | 24.11 | < 0 | < 0 | < 0 | 17.84 |
| MPT-30b-instruct | 36.01 | 84.98 | 233.60 | 132.77 | 76.60 |
| | < 0 | < 0 | < 0 | < 0 | < 0 |
| UL2 | 72.39 | 139.14 | 205.86 | 188.57 | 163.67 |
| | < 0 | < 0 | < 0 | < 0 | < 0 |
| Flan-UL2 | 33.10 | 71.76 | 136.94 | 106.48 | 78.55 |
| | < 0 | < 0 | < 0 | < 0 | < 0 |
| Flan-T5-xxl | 27.07 | 68.02 | 136.47 | 106.08 | 76.69 |
| | < 0 | < 0 | < 0 | < 0 | < 0 |
| ByT5-xxl | 85.83 | 111.53 | 155.27 | 136.99 | 117.80 |
| | < 0 | < 0 | < 0 | < 0 | < 0 |

Table 4: The experimental results on the zero-shot ScrRec using RealtimeQA dataset (upper: ED; lower: RR).

| Model | Ori | 20% | 50% | 100% | KF | KFL | Sub |
|---|---|---|---|---|---|---|---|
| GPT-4 | 97.69 | 97.40 | 96.82 | 93.35 | 93.35 | 95.66 | 62.14 |
| | 100.0 | 99.19 | 97.56 | 87.80 | 87.80 | 94.31 | 0.0 |
| GPT-3.5-turbo | 97.11 | 94.80 | 91.33 | 81.79 | 88.44 | 93.06 | 52.60 |
| | 100.0 | 94.81 | 87.01 | 65.58 | 80.52 | 90.91 | 0.0 |
| text-davinci-003 | 93.93 | 91.62 | 87.28 | 77.46 | 84.10 | 88.15 | 54.91 |
| | 100.0 | 94.07 | 82.96 | 57.78 | 74.81 | 85.19 | 0.0 |
| Falcon-180b | 84.68 | 82.37 | 78.61 | 65.03 | 70.52 | 74.28 | 57.23 |
| | 100.0 | 91.58 | 77.89 | 28.42 | 48.42 | 62.11 | 0.0 |
| Falcon-180b-chat | 75.43 | 72.54 | 71.39 | 58.09 | 63.87 | 67.34 | 54.62 |
| | 100.0 | 86.11 | 80.56 | 16.67 | 44.44 | 61.11 | 0.0 |
| Falcon-40b | 58.96 | 59.25 | 54.62 | 44.51 | 44.51 | 50.29 | 39.60 |
| | 100.0 | 101.49 | 77.61 | 25.37 | 25.37 | 55.22 | 0.0 |
| Falcon-40b-instruct | 65.90 | 63.87 | 61.27 | 48.55 | 53.47 | 60.69 | 46.82 |
| | 100.0 | 89.39 | 75.76 | 9.09 | 34.85 | 72.73 | 0.0 |
| Llama-2-70b | 88.44 | 86.13 | 80.35 | 67.92 | 75.43 | 80.64 | 57.23 |
| | 100.0 | 92.59 | 74.07 | 34.26 | 58.33 | 75.00 | 0.0 |
| Llama-2-70b-chat | 96.24 | 92.49 | 83.82 | 64.74 | 75.43 | 82.66 | 54.91 |
| | 100.0 | 90.91 | 69.93 | 23.78 | 49.65 | 67.13 | 0.0 |
| Llama-2-13b | 66.18 | 65.03 | 57.51 | 47.40 | 52.89 | 57.80 | 44.80 |
| | 100.0 | 94.59 | 59.46 | 12.16 | 37.84 | 60.81 | 0.0 |
| Llama-2-13b-chat | 84.68 | 78.90 | 66.18 | 47.98 | 55.20 | 59.83 | 44.51 |
| | 100.0 | 85.61 | 53.96 | 8.63 | 26.62 | 38.13 | 0.0 |
| Llama-2-7b | 47.40 | 45.09 | 42.77 | 38.73 | 40.75 | 43.64 | 39.31 |
| | 100.0 | 71.43 | 42.86 | < 0 | 17.86 | 53.57 | 0.0 |
| Llama-2-7b-chat | 82.08 | 75.14 | 64.16 | 45.09 | 52.31 | 56.94 | 43.93 |
| | 100.0 | 81.82 | 53.03 | 3.03 | 21.97 | 34.09 | 0.0 |
| MPT-30b | 62.72 | 56.36 | 50.87 | 37.86 | 47.69 | 53.47 | 37.28 |
| | 100.0 | 75.00 | 53.41 | 2.27 | 40.91 | 63.64 | 0.0 |
| MPT-30b-instruct | 65.03 | 66.76 | 58.67 | 47.11 | 54.62 | 61.27 | 39.88 |
| | 100.0 | 106.90 | 74.71 | 28.74 | 58.62 | 85.06 | 0.0 |
| UL2 | 87.57 | 84.68 | 76.30 | 45.95 | 60.69 | 71.10 | 38.44 |
| | 100.0 | 94.12 | 77.06 | 15.29 | 45.29 | 66.47 | 0.0 |
| Flan-UL2 | 98.55 | 95.95 | 89.31 | 49.71 | 65.61 | 76.88 | 46.24 |
| | 100.0 | 95.03 | 82.32 | 6.63 | 37.02 | 58.56 | 0.0 |
| Flan-T5-xxl | 97.40 | 96.24 | 87.86 | 49.42 | 66.47 | 82.66 | 40.75 |
| | 100.0 | 97.96 | 83.16 | 15.31 | 45.41 | 73.98 | 0.0 |
| ByT5-xxl | 73.12 | 57.23 | 46.24 | 26.30 | 35.84 | 41.91 | 25.14 |
| | 100.0 | 66.87 | 43.98 | 2.41 | 22.29 | 34.94 | 0.0 |

Table 5: The experimental results on the zero-shot ScrQA using RealtimeQA dataset (upper: Acc; lower: RPG).

| Model | Ori | 20% | 50% | 100% | KF | KFL | Sub |
|---|---|---|---|---|---|---|---|
| GPT-4 | 96.68 | 96.29 | 94.43 | 87.40 | 93.65 | 96.39 | 46.48 |
| | 100.0 | 99.22 | 95.53 | 81.52 | 93.97 | 99.42 | 0.0 |
| GPT-3.5-turbo | 92.48 | 87.30 | 77.83 | 45.61 | 71.78 | 83.89 | 26.07 |
| | 100.0 | 92.21 | 77.94 | 29.41 | 68.82 | 87.06 | 0.0 |
| text-davinci-003 | 92.29 | 89.45 | 80.27 | 52.44 | 76.66 | 87.50 | 43.85 |
| | 100.0 | 94.15 | 75.20 | 17.74 | 67.74 | 90.12 | 0.0 |
| Falcon-180b | 93.65 | 89.65 | 82.23 | 55.37 | 75.39 | 84.47 | 51.17 |
| | 100.0 | 90.57 | 73.10 | 9.89 | 57.01 | 78.39 | 0.0 |
| Falcon-180b-chat | 92.68 | 89.16 | 78.32 | 49.90 | 70.51 | 80.76 | 45.31 |
| | 100.0 | 92.58 | 69.69 | 9.69 | 53.20 | 74.85 | 0.0 |
| Falcon-40b | 70.12 | 65.04 | 56.45 | 44.34 | 50.78 | 58.30 | 43.26 |
| | 100.0 | 81.09 | 49.09 | 4.00 | 28.00 | 56.00 | 0.0 |
| Falcon-40b-instruct | 75.20 | 67.48 | 58.59 | 43.65 | 50.00 | 60.06 | 42.87 |
| | 100.0 | 76.13 | 48.64 | 2.42 | 22.05 | 53.17 | 0.0 |
| Llama-2-70b | 93.26 | 89.75 | 80.96 | 54.30 | 73.34 | 84.38 | 47.85 |
| | 100.0 | 92.26 | 72.90 | 14.19 | 56.13 | 80.43 | 0.0 |
| Llama-2-70b-chat | 91.11 | 85.74 | 75.10 | 48.24 | 59.67 | 74.02 | 45.12 |
| | 100.0 | 88.32 | 65.18 | 6.79 | 31.63 | 62.85 | 0.0 |
| Llama-2-13b | 69.53 | 63.87 | 53.32 | 38.67 | 44.63 | 52.25 | 37.21 |
| | 100.0 | 82.48 | 49.85 | 4.53 | 22.96 | 46.53 | 0.0 |
| Llama-2-13b-chat | 78.61 | 71.78 | 59.77 | 41.80 | 48.44 | 57.42 | 40.92 |
| | 100.0 | 81.87 | 50.00 | 2.33 | 19.95 | 43.78 | 0.0 |
| Llama-2-7b | 60.55 | 55.08 | 47.46 | 37.60 | 43.07 | 46.39 | 37.30 |
| | 100.0 | 76.47 | 43.70 | 1.26 | 24.79 | 39.08 | 0.0 |
| Llama-2-7b-chat | 73.63 | 64.45 | 56.25 | 40.04 | 45.41 | 53.81 | 39.84 |
| | 100.0 | 72.83 | 48.55 | 0.58 | 16.47 | 41.33 | 0.0 |
| MPT-30b | 74.02 | 69.24 | 59.47 | 47.17 | 50.00 | 57.71 | 43.46 |
| | 100.0 | 84.35 | 52.40 | 12.14 | 21.41 | 46.65 | 0.0 |
| MPT-30b-instruct | 83.50 | 75.68 | 65.43 | 44.82 | 54.30 | 60.64 | 41.41 |
| | 100.0 | 81.44 | 57.08 | 8.12 | 30.63 | 45.71 | 0.0 |
| UL2 | 43.36 | 39.36 | 29.88 | 14.55 | 17.38 | 25.20 | 12.79 |
| | 100.0 | 86.90 | 55.91 | 5.75 | 15.02 | 40.58 | 0.0 |
| Flan-UL2 | 94.24 | 91.21 | 80.96 | 49.80 | 62.99 | 75.39 | 48.14 |
| | 100.0 | 93.43 | 71.19 | 3.60 | 32.20 | 59.11 | 0.0 |
| Flan-T5-xxl | 95.02 | 92.87 | 84.67 | 51.17 | 67.09 | 81.54 | 51.07 |
| | 100.0 | 95.11 | 76.44 | 0.22 | 36.44 | 69.33 | 0.0 |
| ByT5-xxl | 43.16 | 40.23 | 36.23 | 31.93 | 33.50 | 36.62 | 31.84 |
| | 100.0 | 74.14 | 38.79 | 0.86 | 14.66 | 42.24 | 0.0 |

Table 6: The experimental results on the zero-shot ScrQA using DREAM dataset (upper: Acc; lower: RPG).

| Model | Type | Ori | 20% | 50% | 100% | KF | KFL | Sub |
|---|---|---|---|---|---|---|---|---|
| GPT-4 | all | 96.68 | 96.29 | 94.43 | 87.40 | 93.65 | 96.39 | 46.48 |
| | arithmetic | 76.09 | 78.26 | 67.39 | 45.65 | 58.70 | 71.74 | 32.61 |
| | commonsense | 94.69 | 94.41 | 93.02 | 82.40 | 91.90 | 94.41 | 42.46 |
| | logic | 95.97 | 95.84 | 93.42 | 85.77 | 92.62 | 95.97 | 47.65 |
| | matching | 99.07 | 97.22 | 98.15 | 92.59 | 97.22 | 99.07 | 47.22 |
| | summary | 98.47 | 97.71 | 96.18 | 90.08 | 95.42 | 96.95 | 36.64 |
| GPT-3.5-turbo | all | 92.48 | 87.30 | 77.83 | 45.61 | 71.78 | 83.89 | 26.07 |
| | arithmetic | 60.87 | 60.87 | 52.17 | 21.74 | 41.30 | 50.00 | 15.22 |
| | commonsense | 89.11 | 84.36 | 76.54 | 39.39 | 63.69 | 79.61 | 25.42 |
| | logic | 91.01 | 85.64 | 76.11 | 44.83 | 71.28 | 82.95 | 25.10 |
| | matching | 99.07 | 91.67 | 81.48 | 56.48 | 78.70 | 89.81 | 30.56 |
| | summary | 95.42 | 93.13 | 82.44 | 41.22 | 70.99 | 83.97 | 24.43 |
| text-davinci-003 | all | 92.29 | 89.45 | 80.27 | 52.44 | 76.66 | 87.50 | 43.85 |
| | arithmetic | 47.83 | 45.65 | 41.30 | 21.74 | 41.30 | 45.65 | 17.39 |
| | commonsense | 90.78 | 87.99 | 77.37 | 46.37 | 71.79 | 84.92 | 45.25 |
| | logic | 91.01 | 87.92 | 78.39 | 52.89 | 75.03 | 86.71 | 41.34 |
| | matching | 99.07 | 94.44 | 86.11 | 57.41 | 87.96 | 93.52 | 48.15 |
| | summary | 94.66 | 94.66 | 86.26 | 48.09 | 77.86 | 87.79 | 49.62 |
| Falcon-180b | all | 93.65 | 89.65 | 82.23 | 55.37 | 75.39 | 84.47 | 51.17 |
| | arithmetic | 52.17 | 47.83 | 45.65 | 34.78 | 43.48 | 47.83 | 34.78 |
| | commonsense | 91.34 | 88.55 | 79.33 | 56.15 | 74.58 | 83.24 | 52.23 |
| | logic | 92.62 | 88.59 | 79.87 | 52.35 | 73.02 | 82.55 | 50.47 |
| | matching | 98.15 | 90.74 | 87.04 | 62.96 | 78.70 | 89.81 | 51.85 |
| | summary | 96.18 | 93.89 | 89.31 | 67.18 | 87.02 | 91.60 | 56.49 |
| Llama-2-70b | all | 93.26 | 89.75 | 80.96 | 54.30 | 73.34 | 84.38 | 47.85 |
| | arithmetic | 58.70 | 47.83 | 50.00 | 36.96 | 47.83 | 43.48 | 36.96 |
| | commonsense | 91.34 | 89.39 | 80.73 | 53.63 | 73.18 | 84.36 | 44.97 |
| | logic | 92.62 | 88.99 | 79.73 | 54.50 | 73.02 | 84.16 | 48.05 |
| | matching | 94.44 | 89.81 | 79.63 | 50.93 | 73.15 | 79.63 | 50.93 |
| | summary | 96.18 | 94.66 | 86.26 | 58.02 | 79.39 | 90.84 | 50.38 |

Table 7: The experimental results of different categories on the zero-shot ScrQA using DREAM dataset (Acc).

| Model | Type | Ori | 20% | 50% | 100% | KF | KFL | Sub |
|---|---|---|---|---|---|---|---|---|
| GPT-4 | all | 100.0 | 99.22 | 95.53 | 81.52 | 93.97 | 99.42 | 0.0 |
| | arithmetic | 100.0 | 105.00 | 80.00 | 30.00 | 60.00 | 90.00 | 0.0 |
| | commonsense | 100.0 | 99.47 | 96.79 | 76.47 | 94.65 | 99.47 | 0.0 |
| | logic | 100.0 | 99.72 | 94.72 | 78.89 | 93.06 | 100.00 | 0.0 |
| | matching | 100.0 | 96.43 | 98.21 | 87.50 | 96.43 | 100.00 | 0.0 |
| | summary | 100.0 | 98.77 | 96.30 | 86.42 | 95.06 | 97.53 | 0.0 |
| GPT-3.5-turbo | all | 100.0 | 92.21 | 77.94 | 29.41 | 68.82 | 87.06 | 0.0 |
| | arithmetic | 100.0 | 100.00 | 80.95 | 14.29 | 57.14 | 76.19 | 0.0 |
| | commonsense | 100.0 | 92.54 | 80.26 | 21.93 | 60.09 | 85.09 | 0.0 |
| | logic | 100.0 | 91.85 | 77.39 | 29.94 | 70.06 | 87.78 | 0.0 |
| | matching | 100.0 | 89.19 | 74.32 | 37.84 | 70.27 | 86.49 | 0.0 |
| | summary | 100.0 | 96.77 | 81.72 | 23.66 | 65.59 | 83.87 | 0.0 |
| text-davinci-003 | all | 100.0 | 94.15 | 75.20 | 17.74 | 67.74 | 90.12 | 0.0 |
| | arithmetic | 100.0 | 92.86 | 78.57 | 14.29 | 78.57 | 92.86 | 0.0 |
| | commonsense | 100.0 | 93.87 | 70.55 | 2.45 | 58.28 | 87.12 | 0.0 |
| | logic | 100.0 | 93.78 | 74.59 | 23.24 | 67.84 | 91.35 | 0.0 |
| | matching | 100.0 | 90.91 | 74.55 | 18.18 | 78.18 | 89.09 | 0.0 |
| | summary | 100.0 | 100.00 | 81.36 | < 0 | 62.71 | 84.75 | 0.0 |
| Falcon-180b | all | 100.0 | 90.57 | 73.10 | 9.89 | 57.01 | 78.39 | 0.0 |
| | arithmetic | 100.0 | 75.00 | 62.50 | 0.00 | 50.00 | 75.00 | 0.0 |
| | commonsense | 100.0 | 92.86 | 69.29 | 10.00 | 57.14 | 79.29 | 0.0 |
| | logic | 100.0 | 90.45 | 69.75 | 4.46 | 53.50 | 76.11 | 0.0 |
| | matching | 100.0 | 84.00 | 76.00 | 24.00 | 58.00 | 82.00 | 0.0 |
| | summary | 100.0 | 94.23 | 82.69 | 26.92 | 76.92 | 88.46 | 0.0 |
| Llama-2-70b | all | 100.0 | 92.26 | 72.90 | 14.19 | 56.13 | 80.43 | 0.0 |
| | arithmetic | 100.0 | 50.00 | 60.00 | 0.00 | 50.00 | 30.00 | 0.0 |
| | commonsense | 100.0 | 95.78 | 77.11 | 18.67 | 60.84 | 84.94 | 0.0 |
| | logic | 100.0 | 91.87 | 71.08 | 14.46 | 56.02 | 81.02 | 0.0 |
| | matching | 100.0 | 89.36 | 65.96 | 0.00 | 51.06 | 65.96 | 0.0 |
| | summary | 100.0 | 96.67 | 78.33 | 16.67 | 63.33 | 88.33 | 0.0 |

Table 8: The experimental results of different categories on the zero-shot ScrQA using DREAM dataset (RPG).