# OpenReview forum: "Unnatural Error Correction: GPT-4 Can Almost Perfectly Handle Unnatural Scrambled Text"
_EMNLP/2023/Conference — EMNLP 2023 Main_

### Official Review · Reviewer_EzCc · 2023-08-01

**Typos Grammar Style And Presentation Improvements:** "leader" in row 52 should be "reader"
**Soundness:** 4

**Excitement:**

2: Mediocre: This paper makes marginal contributions (vs non-contemporaneous work), so I would rather not see it in the conference.

**Paper Topic And Main Contributions:**

This paper investigates how well LLMs handle the scrambled text. The authors use two evaluation methods: 1. scrambled sentence recovery and 2. scrambled question answering on three OpenAI products: GPT-3.5, ChatGPT and GPT4. They create a dataset which applied different scramble strategies and evaluate the models on this dataset. They show that LLMs can handle the scrambled text.

**Questions For The Authors:**

1. Can you think of any methods to analysis the model behavior using OpenAI API?
2. On application side, can you provide any usage of this scramble recovery ability other than typo correction?

**Reasons To Accept:**

This is an interesting finding on LLMs' capability.

**Reasons To Reject:**

This paper lacks of further analysis and explanation on the LLMs' behavior on scramble texts. E.g. the authors show the differences on tokenization of the regular sentence and scrambled one in Fig 1, but they did not analysis how LLMs encode these different tokens and how LLMs map the scrambled tokens to original ones.
Authors may argue that OpenAI API does not fully support this kind of analysis, which lead to my second point: the authors should also investigate some open-source LLMs like Llama or T5. Although they are not as powerful as OpenAI GPT models, it is easier to conduct more analysis on these models.

**Reproducibility:**

5: Could easily reproduce the results.

**Reviewer Confidence:**

4: Quite sure. I tried to check the important points carefully. It's unlikely, though conceivable, that I missed something that should affect my ratings.

---

> ### Author Rebuttal · Authors · 2023-08-29
>
> Dear Reviewer EzCc:
>
> Thank you very much for your review!
>
> > 1. Can you think of any methods to analysis the model behavior using OpenAI API?
>
> It is difficult to analyze the text generation API of OpenAI. Because we can only get generated contents but nothing about how the model works internally. So, we add some SOTA open-source models in our experiments and do some analysis using Llama-2.
>
> Intuitively, we also thought that LLMs handle scrambled text by mapping the scrambled tokens to original ones. To investigate this, we analyze the hidden state of Llama-2 when inputting scrambled sentences. We calculate the cosine similarity of the hidden state (excluding embedding layers and last two layers, since they are strongly related to input and output, respectively) of each pair of scrambled sentences and original sentences. Then, we average the cosine similarity and normalize it from 0 (sentences with randomly substituted letters) to 1 (original sentences). The results are shown in the table below. For models of all three sizes, the similarity decreases through scrambling severity increases. And the similarity is not related to the performance on tasks. We also sort the samples by recovery rate and separate the samples into two groups based on the correctness on QA tasks and compare their similarities. However, no patterns can be found. So, we assume that LLMs handle scrambled sentences not by mapping them to original ones.
>
> |     | 10%  | 20%  | 30%  | 40%  | 50%  | 60%  | 70%  | 80%  | 90%  | 100% | kf   | kfl  |
> |-----|------|------|------|------|------|------|------|------|------|------|------|------|
> | 70b | 0.91 | 0.81 | 0.72 | 0.63 | 0.54 | 0.46 | 0.37 | 0.29 | 0.23 | 0.15 | 0.27 | 0.41 |
> | 13b | 0.92 | 0.81 |  0.7 | 0.59 | 0.49 | 0.39 |  0.3 | 0.22 | 0.15 | 0.09 | 0.18 | 0.32 |
> | 7b  | 0.97 | 0.92 | 0.86 |  0.8 | 0.73 | 0.67 |  0.6 | 0.53 | 0.46 | 0.39 | 0.49 |  0.6 |
>
> > 2. On application side, can you provide any usage of this scramble recovery ability other than typo correction?
>
> One can require LLMs to do malicious tasks by scrambled text, then the text can pass filters but be still understandable for LLMs (some users already successfully did it and share their trials on some social media). This finding may be helpful to set more effective content filters to prevent malicious use.
>
> This finding shows that the most powerful LLMs can understand and handle scrambled text. Investigating the reasons behind this phenomenon may be challenging, but it could shed light on how LLMs learn and aid in improving their robustness.
>
>
> > The authors should also investigate some open-source LLMs like Llama or T5.
>
> We extend the dataset and investigate the open-source LLMs. The results are shown below:
>
> **Scrambled Sentence Recovery on RealtimeQA (edit distance)**
> |                     |  20percent | 50percent | 100percent | keepfirst | keepfirstlast |
> |---------------------|------------|----------:|-----------:|----------:|--------------:|
> |          gpt-4-0314 |       1.49 |      2.40 |       4.79 |      3.79 |          2.24 |
> |  gpt-3.5-turbo-0301 |       5.80 |     15.38 |      36.87 |     20.76 |          7.40 |
> |    text-davinci-003 |       5.38 |     18.89 |      84.39 |     22.21 |          8.36 |
> |      Llama-2-70b-hf |       7.85 |     37.30 |     169.85 |     56.25 |         18.99 |
> | Llama-2-70b-chat-hf |      11.11 |     54.49 |     196.44 |    106.31 |         43.18 |
> |      Llama-2-13b-hf |      14.18 |     59.13 |     192.67 |    103.50 |         42.67 |
> | Llama-2-13b-chat-hf |      20.75 |     73.04 |     201.50 |    119.52 |         56.03 |
> |       Llama-2-7b-hf |      20.90 |     79.83 |     194.48 |    135.50 |         68.18 |
> |  Llama-2-7b-chat-hf |      25.14 |     83.13 |     196.88 |    132.44 |         73.84 |
> |          falcon-40b |      17.27 |     69.86 |     185.26 |    119.00 |         66.03 |
> | falcon-40b-instruct |      18.79 |     74.26 |     192.86 |    130.82 |         62.39 |
> |             mpt-30b |      14.36 |     54.81 |     181.43 |     90.92 |         36.14 |
> |    mpt-30b-instruct |      19.36 |     66.05 |     170.88 |     99.38 |         50.44 |
> |                 ul2 |      18.12 |     68.79 |     153.53 |    109.47 |         71.70 |
> |            flan-ul2 |      20.51 |     62.91 |     137.15 |    105.67 |         74.35 |
> |         flan-t5-xxl |      16.60 |     59.34 |     136.41 |    105.44 |         73.97 |
> |            byt5-xxl |      79.27 |    104.28 |     148.44 |    123.97 |        102.50 |
>
>
> **Scrambled Sentence Recovery on RealtimeQA(recovery rate)**
> |                     |  20percent | 50percent | 100percent | keepfirst | keepfirstlast |
> |---------------------|------------|----------:|-----------:|----------:|--------------:|
> |          gpt-4-0314 |      94.51 |     96.46 |      96.45 |     96.37 |         97.04 |
> |  gpt-3.5-turbo-0301 |      78.57 |     77.29 |      72.63 |     80.14 |         90.22 |
> |    text-davinci-003 |      80.13 |     72.11 |      37.35 |     78.75 |         88.95 |
> |      Llama-2-70b-hf |       71.0 |     44.93 |         <0 |     46.19 |          74.9 |
> | Llama-2-70b-chat-hf |      58.97 |     19.56 |         <0 |        <0 |         42.93 |
> |      Llama-2-13b-hf |      47.62 |     12.71 |         <0 |      0.99 |         43.59 |
> | Llama-2-13b-chat-hf |      23.35 |        <0 |         <0 |        <0 |         25.93 |
> |       Llama-2-7b-hf |      22.77 |        <0 |         <0 |        <0 |          9.88 |
> |  Llama-2-7b-chat-hf |       7.11 |        <0 |         <0 |        <0 |          2.39 |
> |          falcon-40b |      36.18 |        <0 |         <0 |        <0 |         12.71 |
> | falcon-40b-instruct |      30.56 |        <0 |         <0 |        <0 |         17.53 |
> |             mpt-30b |      46.96 |     19.08 |         <0 |     13.02 |         52.23 |
> |    mpt-30b-instruct |      28.46 |      2.49 |         <0 |      4.92 |         33.33 |
> |                 ul2 |      33.03 |        <0 |         <0 |        <0 |          5.22 |
> |            flan-ul2 |      24.21 |      7.13 |         <0 |        <0 |          1.72 |
> |         flan-t5-xxl |      38.68 |      12.4 |         <0 |        <0 |          2.22 |
> |            byt5-xxl |         <0 |        <0 |         <0 |        <0 |            <0 |
>
>
> **Scrambled QA on RealtimeQA (accuracy)**
> |                     |  original | 20percent | 50percent | 100percent | substituted | keepfirst | keepfirstlast |
> |---------------------|-----------|----------:|----------:|-----------:|------------:|----------:|--------------:|
> |          gpt-4-0314 |     98.18 |     97.26 |     96.05 |      93.01 |       62.92 |     93.92 |         95.74 |
> |  gpt-3.5-turbo-0301 |     97.26 |     96.66 |     91.19 |      84.50 |       55.62 |     86.93 |         91.49 |
> |    text-davinci-003 |     94.22 |     92.10 |     87.84 |      78.72 |       55.62 |     82.98 |         91.49 |
> |      Llama-2-70b-hf |     88.75 |     86.32 |     82.98 |      65.65 |       59.88 |     78.42 |         82.67 |
> | Llama-2-70b-chat-hf |     96.05 |     92.71 |     85.71 |      64.74 |       55.62 |     75.68 |         84.19 |
> |      Llama-2-13b-hf |     66.87 |     62.92 |     57.75 |      46.81 |       46.20 |     55.02 |         55.62 |
> | Llama-2-13b-chat-hf |     85.41 |     75.99 |     66.87 |      48.02 |       45.90 |     55.32 |         61.40 |
> |       Llama-2-7b-hf |     46.81 |     44.38 |     41.64 |      40.73 |       38.30 |     40.12 |         44.07 |
> |  Llama-2-7b-chat-hf |     82.67 |     77.51 |     65.05 |      46.20 |       46.20 |     50.76 |         60.79 |
> |          falcon-40b |     49.85 |     47.42 |     41.03 |      29.79 |       28.88 |     31.00 |         34.65 |
> | falcon-40b-instruct |     57.14 |     56.23 |     48.94 |      35.56 |       33.13 |     39.21 |         41.03 |
> |             mpt-30b |     62.31 |     56.53 |     50.15 |      41.64 |       39.51 |     47.42 |         50.15 |
> |    mpt-30b-instruct |     64.13 |     63.83 |     58.97 |      48.33 |       42.86 |     56.23 |         62.92 |
> |                 ul2 |     87.23 |     84.19 |     75.68 |      48.94 |       38.60 |     62.92 |         70.52 |
> |         flan-t5-xxl |     97.26 |     94.53 |     88.15 |      50.76 |       38.30 |     65.35 |         79.64 |
> |            byt5-xxl |     72.04 |     56.84 |     46.81 |      27.36 |       24.62 |     34.35 |         42.55 |
>
>
> **Scrambled QA on RealtimeQA (relative accuracy gain)**
> |                     |  original | 20percent | 50percent | 100percent | substituted | keepfirst | keepfirstlast |
> |---------------------|-----------|----------:|----------:|-----------:|------------:|----------:|--------------:|
> |          gpt-4-0314 |       1.0 |      0.97 |      0.94 |       0.85 |         0.0 |      0.88 |          0.93 |
> |  gpt-3.5-turbo-0301 |       1.0 |      0.99 |      0.85 |       0.69 |         0.0 |      0.75 |          0.86 |
> |    text-davinci-003 |       1.0 |      0.94 |      0.83 |       0.60 |         0.0 |      0.71 |          0.93 |
> |      Llama-2-70b-hf |       1.0 |      0.92 |      0.80 |       0.20 |         0.0 |      0.64 |          0.79 |
> | Llama-2-70b-chat-hf |       1.0 |      0.92 |      0.74 |       0.23 |         0.0 |      0.50 |          0.71 |
> |      Llama-2-13b-hf |       1.0 |      0.81 |      0.56 |       0.03 |         0.0 |      0.43 |          0.46 |
> | Llama-2-13b-chat-hf |       1.0 |      0.76 |      0.53 |       0.05 |         0.0 |      0.24 |          0.39 |
> |       Llama-2-7b-hf |       1.0 |      0.71 |      0.39 |       0.29 |         0.0 |      0.21 |          0.68 |
> |  Llama-2-7b-chat-hf |       1.0 |      0.86 |      0.52 |       0.00 |         0.0 |      0.13 |          0.40 |
> |          falcon-40b |       1.0 |      0.88 |      0.58 |       0.04 |         0.0 |      0.10 |          0.28 |
> | falcon-40b-instruct |       1.0 |      0.96 |      0.66 |       0.10 |         0.0 |      0.25 |          0.33 |
> |             mpt-30b |       1.0 |      0.75 |      0.47 |       0.09 |         0.0 |      0.35 |          0.47 |
> |    mpt-30b-instruct |       1.0 |      0.99 |      0.76 |       0.26 |         0.0 |      0.63 |          0.94 |
> |                 ul2 |       1.0 |      0.94 |      0.76 |       0.21 |         0.0 |      0.50 |          0.66 |
> |         flan-t5-xxl |       1.0 |      0.95 |      0.85 |       0.21 |         0.0 |      0.46 |          0.70 |
> |            byt5-xxl |       1.0 |      0.68 |      0.47 |       0.06 |         0.0 |      0.21 |          0.38 |
>
>
> **Scrambled QA on DREAM (accuracy)**
> |                     |  original | 20percent | 50percent | 100percent | substituted | keepfirst | keepfirstlast |
> |---------------------|-----------|----------:|----------:|-----------:|------------:|----------:|--------------:|
> |          gpt-4-0314 |     96.68 |     96.29 |     94.73 |      88.48 |       45.70 |     93.95 |         96.48 |
> |  gpt-3.5-turbo-0301 |     94.92 |     90.23 |     81.05 |      44.73 |       21.29 |     73.44 |         85.74 |
> |    text-davinci-003 |     93.36 |     91.02 |     81.84 |      54.69 |       45.12 |     79.30 |         88.48 |
> |      Llama-2-70b-hf |     92.97 |     90.62 |     83.40 |      57.23 |       50.20 |     74.02 |         85.16 |
> | Llama-2-70b-chat-hf |     91.02 |     86.13 |     76.95 |      48.44 |       45.70 |     59.77 |         74.41 |
> |      Llama-2-13b-hf |     70.51 |     65.62 |     54.69 |      40.82 |       38.28 |     42.38 |         52.93 |
> | Llama-2-13b-chat-hf |     80.08 |     74.80 |     62.89 |      42.58 |       41.02 |     48.05 |         58.59 |
> |       Llama-2-7b-hf |     60.74 |     55.66 |     48.83 |      38.48 |       38.67 |     42.97 |         46.29 |
> |  Llama-2-7b-chat-hf |     75.00 |     66.21 |     57.03 |      40.04 |       38.09 |     42.97 |         52.93 |
> |          falcon-40b |     64.06 |     59.77 |     53.52 |      44.34 |       42.77 |     48.63 |         54.69 |
> | falcon-40b-instruct |     73.05 |     65.62 |     57.23 |      42.97 |       43.16 |     48.63 |         57.03 |
> |             mpt-30b |     76.37 |     73.24 |     60.74 |      47.85 |       45.31 |     50.78 |         58.98 |
> |    mpt-30b-instruct |     86.13 |     78.52 |     69.53 |      45.12 |       42.77 |     55.47 |         63.09 |
> |                 ul2 |     44.14 |     41.02 |     33.20 |      20.31 |       17.77 |     22.27 |         28.71 |
> |            flan-ul2 |     94.53 |     91.02 |     82.23 |      53.71 |       49.22 |     65.23 |         76.37 |
> |         flan-t5-xxl |     96.68 |     94.34 |     86.72 |      54.49 |       53.52 |     70.70 |         84.77 |
>
>
> **Scrambled QA on DREAM (relative accuracy gain)**
> |                     |  original | 20percent | 50percent | 100percent | substituted | keepfirst | keepfirstlast |
> |---------------------|-----------|----------:|----------:|-----------:|------------:|----------:|--------------:|
> |          gpt-4-0314 |       1.0 |      0.99 |      0.96 |       0.84 |         0.0 |      0.95 |          1.00 |
> |  gpt-3.5-turbo-0301 |       1.0 |      0.94 |      0.81 |       0.32 |         0.0 |      0.71 |          0.88 |
> |    text-davinci-003 |       1.0 |      0.95 |      0.76 |       0.20 |         0.0 |      0.71 |          0.90 |
> |      Llama-2-70b-hf |       1.0 |      0.95 |      0.78 |       0.16 |         0.0 |      0.56 |          0.82 |
> | Llama-2-70b-chat-hf |       1.0 |      0.89 |      0.69 |       0.06 |         0.0 |      0.31 |          0.63 |
> |      Llama-2-13b-hf |       1.0 |      0.85 |      0.51 |       0.08 |         0.0 |      0.13 |          0.45 |
> | Llama-2-13b-chat-hf |       1.0 |      0.86 |      0.56 |       0.04 |         0.0 |      0.18 |          0.45 |
> |       Llama-2-7b-hf |       1.0 |      0.77 |      0.46 |      -0.01 |         0.0 |      0.19 |          0.35 |
> |  Llama-2-7b-chat-hf |       1.0 |      0.76 |      0.51 |       0.05 |         0.0 |      0.13 |          0.40 |
> |          falcon-40b |       1.0 |      0.80 |      0.50 |       0.07 |         0.0 |      0.28 |          0.56 |
> | falcon-40b-instruct |       1.0 |      0.75 |      0.47 |      -0.01 |         0.0 |      0.18 |          0.46 |
> |             mpt-30b |       1.0 |      0.90 |      0.50 |       0.08 |         0.0 |      0.18 |          0.44 |
> |    mpt-30b-instruct |       1.0 |      0.82 |      0.62 |       0.05 |         0.0 |      0.29 |          0.47 |
> |                 ul2 |       1.0 |      0.88 |      0.59 |       0.10 |         0.0 |      0.17 |          0.41 |
> |            flan-ul2 |       1.0 |      0.92 |      0.73 |       0.10 |         0.0 |      0.35 |          0.60 |
> |         flan-t5-xxl |       1.0 |      0.95 |      0.77 |       0.02 |         0.0 |      0.40 |          0.72 |
>
>
> >  "leader" in row 52 should be "reader"
>
> Thank you so much for your careful inspection! We will definitely correct them in modified paper.

---

### Official Review · Reviewer_Lbf2 · 2023-08-05

**Soundness:** 4

**Excitement:**

4: Strong: This paper deepens the understanding of some phenomenon or lowers the barriers to an existing research direction.

**Paper Topic And Main Contributions:**

In this paper, the authors evaluated the performance of the GPT-4 method for two tasks: scrambled sentence recovery and scrambled question answering. Also, the authors propose two new datasets based on RealtimeQA and AQUA.

The experimental results show that the GPT-4 approach employed in this work achieves better results compared to other LLMs, namely Text-davinci-003 and GPT-3.5-turbo.
This research work is interesting and important.

**Reasons To Accept:**

Strength:

The paper is well-grounded in current literature, citing recent approaches and advances in the field.

The methodology seems simple but well thought-out.

**Reasons To Reject:**

The reviewer has the following comments to improve the quality of the paper:

 I suggest the author add the method diagram to the paper.

 Clearly introduce the weakness of existing approaches in the area of unnatural scrambled text, and emphasize the strength of the proposed method, which narrows the research gap.

 The validation of the method's effectiveness must be more convincing. Please include a comparison with many LLMs state-of-the-art solutions (e.g., Alpaca, FLAN-T5, Llama).

**Reproducibility:**

5: Could easily reproduce the results.

**Reviewer Confidence:**

3: Pretty sure, but there's a chance I missed something. Although I have a good feel for this area in general, I did not carefully check the paper's details, e.g., the math, experimental design, or novelty.

---

> ### Author Rebuttal · Authors · 2023-08-29
>
> Dear Reviewer Lbf2:
>
> Thank you very much for your review!
>
> > Clearly introduce the weakness of existing approaches in the area of unnatural scrambled text, and emphasize the strength of the proposed method, which narrows the research gap.
>
> Thank you for your suggestion. We will definitely add the diagram to clarify the difference from prior works, while we can not include figures during the rebuttal. Namely, we will add a figure comparing three types of robustness against (1) unnatural word-level permutation, (2) natural character-level permutation, and (3) unnatural character-level permutation (proposal). To best of all knowledge, there are no existing studies investigating LLMs' ability to handle the character-level permutation in extreme level that drastically change the tokenization. Here, we add a simple table to demonstrate the differences, we will definitely add the comparison in more details in the modified paper.
>
> |   |                                         Category | Example                                           |    Type   | Severity |
> |---|-------------------------------------------------:|---------------------------------------------------|:---------:|:--------:|
> |   |                        Original (No Permutation) | Does the order of letters in word matter for LLMs? |    None   |   None   |
> |   |     Unnatural Word-level Permutation (e.g., [1]) | Does letters the order for LLMs or word in matter? |    Word   |   High   |
> |   |  Natural Character-level Permutation (e.g., [2]) | Does the odrer of letters in wor matter for LLMs? | Character |    Low   |
> |   | Unnatural Character-level Permutation (proposal) | oeDs eth doerr fo lettres ni rwor ttemar rof LMLs? | Character |   Extreme   |
>
> [1] Sinha, Koustuv, et al. "UnNatural Language Inference." Proceedings of the 59th Annual Meeting of the Association for Computational Linguistics and the 11th International Joint Conference on Natural Language Processing (Volume 1: Long Papers). 2021.
>
> [2] Wang, Jindong, et al. "On the robustness of chatgpt: An adversarial and out-of-distribution perspective." arXiv preprint arXiv:2302.12095 (2023).
>
>
> >  The validation of the method's effectiveness must be more convincing. Please include a comparison with many LLMs state-of-the-art solutions (e.g., Alpaca, FLAN-T5, Llama).
>
>
> Following the reviewers’ comments, we conducted experiments using various open-source LLMs (including llama-2, falcon, mpt, ul2, flan-ul2, flan-t5, and byt5). The model covers diverse model architectures (decoder-only and encoder-decoder), model sizes (from 7b to 70b), and training objectives (e.g., with or without further finetuning) and tokenizers (e.g., tokenizer-free: byt5-xxl). The results are shown below:
>
> **Scrambled Sentence Recovery on RealtimeQA (edit distance)**
> |                     |  20percent | 50percent | 100percent | keepfirst | keepfirstlast |
> |---------------------|------------|----------:|-----------:|----------:|--------------:|
> |          gpt-4-0314 |       1.49 |      2.40 |       4.79 |      3.79 |          2.24 |
> |  gpt-3.5-turbo-0301 |       5.80 |     15.38 |      36.87 |     20.76 |          7.40 |
> |    text-davinci-003 |       5.38 |     18.89 |      84.39 |     22.21 |          8.36 |
> |      Llama-2-70b-hf |       7.85 |     37.30 |     169.85 |     56.25 |         18.99 |
> | Llama-2-70b-chat-hf |      11.11 |     54.49 |     196.44 |    106.31 |         43.18 |
> |      Llama-2-13b-hf |      14.18 |     59.13 |     192.67 |    103.50 |         42.67 |
> | Llama-2-13b-chat-hf |      20.75 |     73.04 |     201.50 |    119.52 |         56.03 |
> |       Llama-2-7b-hf |      20.90 |     79.83 |     194.48 |    135.50 |         68.18 |
> |  Llama-2-7b-chat-hf |      25.14 |     83.13 |     196.88 |    132.44 |         73.84 |
> |          falcon-40b |      17.27 |     69.86 |     185.26 |    119.00 |         66.03 |
> | falcon-40b-instruct |      18.79 |     74.26 |     192.86 |    130.82 |         62.39 |
> |             mpt-30b |      14.36 |     54.81 |     181.43 |     90.92 |         36.14 |
> |    mpt-30b-instruct |      19.36 |     66.05 |     170.88 |     99.38 |         50.44 |
> |                 ul2 |      18.12 |     68.79 |     153.53 |    109.47 |         71.70 |
> |            flan-ul2 |      20.51 |     62.91 |     137.15 |    105.67 |         74.35 |
> |         flan-t5-xxl |      16.60 |     59.34 |     136.41 |    105.44 |         73.97 |
> |            byt5-xxl |      79.27 |    104.28 |     148.44 |    123.97 |        102.50 |
>
>
> **Scrambled Sentence Recovery on RealtimeQA(recovery rate)**
> |                     |  20percent | 50percent | 100percent | keepfirst | keepfirstlast |
> |---------------------|------------|----------:|-----------:|----------:|--------------:|
> |          gpt-4-0314 |      94.51 |     96.46 |      96.45 |     96.37 |         97.04 |
> |  gpt-3.5-turbo-0301 |      78.57 |     77.29 |      72.63 |     80.14 |         90.22 |
> |    text-davinci-003 |      80.13 |     72.11 |      37.35 |     78.75 |         88.95 |
> |      Llama-2-70b-hf |       71.0 |     44.93 |         <0 |     46.19 |          74.9 |
> | Llama-2-70b-chat-hf |      58.97 |     19.56 |         <0 |        <0 |         42.93 |
> |      Llama-2-13b-hf |      47.62 |     12.71 |         <0 |      0.99 |         43.59 |
> | Llama-2-13b-chat-hf |      23.35 |        <0 |         <0 |        <0 |         25.93 |
> |       Llama-2-7b-hf |      22.77 |        <0 |         <0 |        <0 |          9.88 |
> |  Llama-2-7b-chat-hf |       7.11 |        <0 |         <0 |        <0 |          2.39 |
> |          falcon-40b |      36.18 |        <0 |         <0 |        <0 |         12.71 |
> | falcon-40b-instruct |      30.56 |        <0 |         <0 |        <0 |         17.53 |
> |             mpt-30b |      46.96 |     19.08 |         <0 |     13.02 |         52.23 |
> |    mpt-30b-instruct |      28.46 |      2.49 |         <0 |      4.92 |         33.33 |
> |                 ul2 |      33.03 |        <0 |         <0 |        <0 |          5.22 |
> |            flan-ul2 |      24.21 |      7.13 |         <0 |        <0 |          1.72 |
> |         flan-t5-xxl |      38.68 |      12.4 |         <0 |        <0 |          2.22 |
> |            byt5-xxl |         <0 |        <0 |         <0 |        <0 |            <0 |
>
>
> **Scrambled QA on RealtimeQA (accuracy)**
> |                     |  original | 20percent | 50percent | 100percent | substituted | keepfirst | keepfirstlast |
> |---------------------|-----------|----------:|----------:|-----------:|------------:|----------:|--------------:|
> |          gpt-4-0314 |     98.18 |     97.26 |     96.05 |      93.01 |       62.92 |     93.92 |         95.74 |
> |  gpt-3.5-turbo-0301 |     97.26 |     96.66 |     91.19 |      84.50 |       55.62 |     86.93 |         91.49 |
> |    text-davinci-003 |     94.22 |     92.10 |     87.84 |      78.72 |       55.62 |     82.98 |         91.49 |
> |      Llama-2-70b-hf |     88.75 |     86.32 |     82.98 |      65.65 |       59.88 |     78.42 |         82.67 |
> | Llama-2-70b-chat-hf |     96.05 |     92.71 |     85.71 |      64.74 |       55.62 |     75.68 |         84.19 |
> |      Llama-2-13b-hf |     66.87 |     62.92 |     57.75 |      46.81 |       46.20 |     55.02 |         55.62 |
> | Llama-2-13b-chat-hf |     85.41 |     75.99 |     66.87 |      48.02 |       45.90 |     55.32 |         61.40 |
> |       Llama-2-7b-hf |     46.81 |     44.38 |     41.64 |      40.73 |       38.30 |     40.12 |         44.07 |
> |  Llama-2-7b-chat-hf |     82.67 |     77.51 |     65.05 |      46.20 |       46.20 |     50.76 |         60.79 |
> |          falcon-40b |     49.85 |     47.42 |     41.03 |      29.79 |       28.88 |     31.00 |         34.65 |
> | falcon-40b-instruct |     57.14 |     56.23 |     48.94 |      35.56 |       33.13 |     39.21 |         41.03 |
> |             mpt-30b |     62.31 |     56.53 |     50.15 |      41.64 |       39.51 |     47.42 |         50.15 |
> |    mpt-30b-instruct |     64.13 |     63.83 |     58.97 |      48.33 |       42.86 |     56.23 |         62.92 |
> |                 ul2 |     87.23 |     84.19 |     75.68 |      48.94 |       38.60 |     62.92 |         70.52 |
> |         flan-t5-xxl |     97.26 |     94.53 |     88.15 |      50.76 |       38.30 |     65.35 |         79.64 |
> |            byt5-xxl |     72.04 |     56.84 |     46.81 |      27.36 |       24.62 |     34.35 |         42.55 |
>
>
> **Scrambled QA on RealtimeQA (relative accuracy gain)**
> |                     |  original | 20percent | 50percent | 100percent | substituted | keepfirst | keepfirstlast |
> |---------------------|-----------|----------:|----------:|-----------:|------------:|----------:|--------------:|
> |          gpt-4-0314 |       1.0 |      0.97 |      0.94 |       0.85 |         0.0 |      0.88 |          0.93 |
> |  gpt-3.5-turbo-0301 |       1.0 |      0.99 |      0.85 |       0.69 |         0.0 |      0.75 |          0.86 |
> |    text-davinci-003 |       1.0 |      0.94 |      0.83 |       0.60 |         0.0 |      0.71 |          0.93 |
> |      Llama-2-70b-hf |       1.0 |      0.92 |      0.80 |       0.20 |         0.0 |      0.64 |          0.79 |
> | Llama-2-70b-chat-hf |       1.0 |      0.92 |      0.74 |       0.23 |         0.0 |      0.50 |          0.71 |
> |      Llama-2-13b-hf |       1.0 |      0.81 |      0.56 |       0.03 |         0.0 |      0.43 |          0.46 |
> | Llama-2-13b-chat-hf |       1.0 |      0.76 |      0.53 |       0.05 |         0.0 |      0.24 |          0.39 |
> |       Llama-2-7b-hf |       1.0 |      0.71 |      0.39 |       0.29 |         0.0 |      0.21 |          0.68 |
> |  Llama-2-7b-chat-hf |       1.0 |      0.86 |      0.52 |       0.00 |         0.0 |      0.13 |          0.40 |
> |          falcon-40b |       1.0 |      0.88 |      0.58 |       0.04 |         0.0 |      0.10 |          0.28 |
> | falcon-40b-instruct |       1.0 |      0.96 |      0.66 |       0.10 |         0.0 |      0.25 |          0.33 |
> |             mpt-30b |       1.0 |      0.75 |      0.47 |       0.09 |         0.0 |      0.35 |          0.47 |
> |    mpt-30b-instruct |       1.0 |      0.99 |      0.76 |       0.26 |         0.0 |      0.63 |          0.94 |
> |                 ul2 |       1.0 |      0.94 |      0.76 |       0.21 |         0.0 |      0.50 |          0.66 |
> |         flan-t5-xxl |       1.0 |      0.95 |      0.85 |       0.21 |         0.0 |      0.46 |          0.70 |
> |            byt5-xxl |       1.0 |      0.68 |      0.47 |       0.06 |         0.0 |      0.21 |          0.38 |
>
>
> **Scrambled QA on DREAM (accuracy)**
> |                     |  original | 20percent | 50percent | 100percent | substituted | keepfirst | keepfirstlast |
> |---------------------|-----------|----------:|----------:|-----------:|------------:|----------:|--------------:|
> |          gpt-4-0314 |     96.68 |     96.29 |     94.73 |      88.48 |       45.70 |     93.95 |         96.48 |
> |  gpt-3.5-turbo-0301 |     94.92 |     90.23 |     81.05 |      44.73 |       21.29 |     73.44 |         85.74 |
> |    text-davinci-003 |     93.36 |     91.02 |     81.84 |      54.69 |       45.12 |     79.30 |         88.48 |
> |      Llama-2-70b-hf |     92.97 |     90.62 |     83.40 |      57.23 |       50.20 |     74.02 |         85.16 |
> | Llama-2-70b-chat-hf |     91.02 |     86.13 |     76.95 |      48.44 |       45.70 |     59.77 |         74.41 |
> |      Llama-2-13b-hf |     70.51 |     65.62 |     54.69 |      40.82 |       38.28 |     42.38 |         52.93 |
> | Llama-2-13b-chat-hf |     80.08 |     74.80 |     62.89 |      42.58 |       41.02 |     48.05 |         58.59 |
> |       Llama-2-7b-hf |     60.74 |     55.66 |     48.83 |      38.48 |       38.67 |     42.97 |         46.29 |
> |  Llama-2-7b-chat-hf |     75.00 |     66.21 |     57.03 |      40.04 |       38.09 |     42.97 |         52.93 |
> |          falcon-40b |     64.06 |     59.77 |     53.52 |      44.34 |       42.77 |     48.63 |         54.69 |
> | falcon-40b-instruct |     73.05 |     65.62 |     57.23 |      42.97 |       43.16 |     48.63 |         57.03 |
> |             mpt-30b |     76.37 |     73.24 |     60.74 |      47.85 |       45.31 |     50.78 |         58.98 |
> |    mpt-30b-instruct |     86.13 |     78.52 |     69.53 |      45.12 |       42.77 |     55.47 |         63.09 |
> |                 ul2 |     44.14 |     41.02 |     33.20 |      20.31 |       17.77 |     22.27 |         28.71 |
> |            flan-ul2 |     94.53 |     91.02 |     82.23 |      53.71 |       49.22 |     65.23 |         76.37 |
> |         flan-t5-xxl |     96.68 |     94.34 |     86.72 |      54.49 |       53.52 |     70.70 |         84.77 |
>
>
> **Scrambled QA on DREAM (relative accuracy gain)**
> |                     |  original | 20percent | 50percent | 100percent | substituted | keepfirst | keepfirstlast |
> |---------------------|-----------|----------:|----------:|-----------:|------------:|----------:|--------------:|
> |          gpt-4-0314 |       1.0 |      0.99 |      0.96 |       0.84 |         0.0 |      0.95 |          1.00 |
> |  gpt-3.5-turbo-0301 |       1.0 |      0.94 |      0.81 |       0.32 |         0.0 |      0.71 |          0.88 |
> |    text-davinci-003 |       1.0 |      0.95 |      0.76 |       0.20 |         0.0 |      0.71 |          0.90 |
> |      Llama-2-70b-hf |       1.0 |      0.95 |      0.78 |       0.16 |         0.0 |      0.56 |          0.82 |
> | Llama-2-70b-chat-hf |       1.0 |      0.89 |      0.69 |       0.06 |         0.0 |      0.31 |          0.63 |
> |      Llama-2-13b-hf |       1.0 |      0.85 |      0.51 |       0.08 |         0.0 |      0.13 |          0.45 |
> | Llama-2-13b-chat-hf |       1.0 |      0.86 |      0.56 |       0.04 |         0.0 |      0.18 |          0.45 |
> |       Llama-2-7b-hf |       1.0 |      0.77 |      0.46 |      -0.01 |         0.0 |      0.19 |          0.35 |
> |  Llama-2-7b-chat-hf |       1.0 |      0.76 |      0.51 |       0.05 |         0.0 |      0.13 |          0.40 |
> |          falcon-40b |       1.0 |      0.80 |      0.50 |       0.07 |         0.0 |      0.28 |          0.56 |
> | falcon-40b-instruct |       1.0 |      0.75 |      0.47 |      -0.01 |         0.0 |      0.18 |          0.46 |
> |             mpt-30b |       1.0 |      0.90 |      0.50 |       0.08 |         0.0 |      0.18 |          0.44 |
> |    mpt-30b-instruct |       1.0 |      0.82 |      0.62 |       0.05 |         0.0 |      0.29 |          0.47 |
> |                 ul2 |       1.0 |      0.88 |      0.59 |       0.10 |         0.0 |      0.17 |          0.41 |
> |            flan-ul2 |       1.0 |      0.92 |      0.73 |       0.10 |         0.0 |      0.35 |          0.60 |
> |         flan-t5-xxl |       1.0 |      0.95 |      0.77 |       0.02 |         0.0 |      0.40 |          0.72 |

---

### Official Review · Reviewer_tGXe · 2023-08-08

**Soundness:** 4

**Excitement:**

4: Strong: This paper deepens the understanding of some phenomenon or lowers the barriers to an existing research direction.

**Paper Topic And Main Contributions:**

This work probes the robustness of recent language models, including GPT-3, GPT-3.5 and GPT-4 to scrambled text (with character-level permutations). This is done by addressing two different tasks: 1) scrambled sentence recovery and 2) scrambled question answering and by constructing test sets for each of the tasks. The results show an impressive performance for GPT-4 that succeeds almost perfectly on the test sets, also in case where the scrambling is unnatural i.e. the first and last letters of the word are also scrambled.

**Reasons To Accept:**

- The paper addresses an interesting research question. Such an investigation could contribute to the probing of the abilities of recent large language models and to the comparison between different language models.
- The research question is addressed from diverse perspectives, in particular considering two different tasks, experimenting on multiple corpora and using multiple evaluation metrics.

**Reasons To Reject:**

My main concern is that the test sets (179 samples and 155 samples for the scrambled sentence recovery task and the scrambled question answering task respectively) are too small to draw reliable conclusions. Similarly, the experiments on an additional dataset for the scrambled question answering task only concern 100 samples.

**Reproducibility:**

4: Could mostly reproduce the results, but there may be some variation because of sample variance or minor variations in their interpretation of the protocol or method.

**Reviewer Confidence:**

4: Quite sure. I tried to check the important points carefully. It's unlikely, though conceivable, that I missed something that should affect my ratings.

**Typos Grammar Style And Presentation Improvements:**

Typos, Grammar, Style:
The writing of the paper should considerably be improved and grammatical errors should be corrected.

- lines 014-015: the human-like abilities -> human-like abilities
- line 024: the 95% of original words -> 95% of the original words
- line 027: tend to emergent -> tend to emerge
- line 031: has been proposed -> have been proposed
- line 034-035: are still under mysteries -> are still a mystery
- line 042-043: on the level of unnatural for human intuition -> on an unnatural level for human intuition
- line 044: it challenges -> they challenge
- line 052: As leader may ware -> Note that
- line 059: help to analyze -> help one to analyze
- lines 054-055: humans often can easily recognize -> humans can often easily recognize
- line 063: test suite -> a test suite
- line 071: change -> changes
- lines 099-100: LLMs do not understand what should do -> LLMa do not "understand" what they should do
- line 101:  sentence -> the sentence
- line 106: scrambled version -> a scrambled version
- line 142: keep invariant -> are kept invariant
- line 185: be appeared -> appear
- lines 197- 198: of zero-shot setting -> on a zero-shot setting
- line 200: type -> types
- line 207: number -> numbers
- line 210: decrease -> decreases
- lines 210-211: getting become difficult -> become difficult (or get difficult)
- line 212: keeps -> remains
- line 213: regardless -> regardless of
- line 216: A comma should be added between "setting" and "compared"
- lines 217-218: including different evaluation metric -> also with a different evaluation metric
- lines 220-221: severity -> severities
- line 225: increase -> increases
- line 227: do -> does
- line 232: plot -> plots
- line 235: its -> of its
- line 255: reconstruct -> reconstructs
- line 256: answer -> answers
- lines 259 and 296:  the comma after "that" should be removed
- line 263: and provide analysis -> and an analysis
- line 271: change - > changes
- line 297: and easier -> and it is easy
- line 308: And We -> We
- line 310: of the models ->  about the models
- line 312: superior -> is superior
- line 314: severer -> more severe
- lines 315-316: belong to -> is due to

Presentation:
- In the abstract, two different fonts are used for GPT-4. I think the one in line 019 should be kept since it is also used in the rest of the paper.
- Section 2:  I think it should be split into two separate sections: Related Work and then Scrambled Bench
- lines 148-154: the motivation for the new measure and its interpretation should be provided.
- lies 158-165:  Is RPG a new measure? The motivation for it and its interpretation should be provided.
- lines 201-204: it would be better to write the location of the tasks on the figure in parenthesis *after* rather than *before* the name of the
  task. For example: on (left) zero-shot scramble sentence recovery -> on zero-shot scramble sentence recovery (left).
- line 217: It should be specified that Tables 3 and 4 are in the Appendix. Also, I don't see the edit distance metric in these tables.
- line 240: Results 3: it should be made clear that you focus here on the scrambled QA task.
- lines 298-299: This should be put in a footnote, preferably in the abstract so it will appear on the first page. On the final version, you could put the link to the code instead.

---

> ### Author Rebuttal · Authors · 2023-08-29
>
> Dear Reviewer tGXe:
>
> Thank you very much for your review!
>
> > My main concern is that the test sets (179 samples and 155 samples for the scrambled sentence recovery task and the scrambled question answering task respectively) are too small to draw reliable conclusions. Similarly, the experiments on an additional dataset for the scrambled question answering task only concern 100 samples.
>
> We extend the scrambled realtimeqa dataset (using data from 2023/03/17 to 2023/08/04). The data sizes are extended to 408 and 329 for the scrambled sentence recovery task and the scrambled question answering task, respectively. We would like to evaluate open-source models on this task, but the additional dataset AQUA itself is too challenging to some open-source models. Instead, we process DREAM dataset for a dialogue comprehension task as another additional dataset for question answering task (512 samples with categorical labels are selected). The setting of DREAM is similar to the scrambled RealtimeQA dataset (ask LLMs to answer questions based on scrambled dialogues). Furthermore, for fairer comparisons, we set the results of context with randomly substituted letters to replace that of no context as the lower bound. Additionally, we also test open-source models (Llama-2 series, T5 series, Falcon series and MPT series) on these tasks.
> In the results, the results of GPT models are similar to that we reported in the paper. GPT-4 can outperform all other models significantly, especially on the extreme situations. For open-source models, several ones (e.g., Llama-2-70b) can handle scrambled text of mild level. No open-source models can well handle the 100% scrambled text.
>
> **Scrambled Sentence Recovery on RealtimeQA (edit distance)**
> |                     |  20percent | 50percent | 100percent | keepfirst | keepfirstlast |
> |---------------------|------------|----------:|-----------:|----------:|--------------:|
> |          gpt-4-0314 |       1.49 |      2.40 |       4.79 |      3.79 |          2.24 |
> |  gpt-3.5-turbo-0301 |       5.80 |     15.38 |      36.87 |     20.76 |          7.40 |
> |    text-davinci-003 |       5.38 |     18.89 |      84.39 |     22.21 |          8.36 |
> |      Llama-2-70b-hf |       7.85 |     37.30 |     169.85 |     56.25 |         18.99 |
> | Llama-2-70b-chat-hf |      11.11 |     54.49 |     196.44 |    106.31 |         43.18 |
> |      Llama-2-13b-hf |      14.18 |     59.13 |     192.67 |    103.50 |         42.67 |
> | Llama-2-13b-chat-hf |      20.75 |     73.04 |     201.50 |    119.52 |         56.03 |
> |       Llama-2-7b-hf |      20.90 |     79.83 |     194.48 |    135.50 |         68.18 |
> |  Llama-2-7b-chat-hf |      25.14 |     83.13 |     196.88 |    132.44 |         73.84 |
> |          falcon-40b |      17.27 |     69.86 |     185.26 |    119.00 |         66.03 |
> | falcon-40b-instruct |      18.79 |     74.26 |     192.86 |    130.82 |         62.39 |
> |             mpt-30b |      14.36 |     54.81 |     181.43 |     90.92 |         36.14 |
> |    mpt-30b-instruct |      19.36 |     66.05 |     170.88 |     99.38 |         50.44 |
> |                 ul2 |      18.12 |     68.79 |     153.53 |    109.47 |         71.70 |
> |            flan-ul2 |      20.51 |     62.91 |     137.15 |    105.67 |         74.35 |
> |         flan-t5-xxl |      16.60 |     59.34 |     136.41 |    105.44 |         73.97 |
> |            byt5-xxl |      79.27 |    104.28 |     148.44 |    123.97 |        102.50 |
>
>
> **Scrambled Sentence Recovery on RealtimeQA(recovery rate)**
> |                     |  20percent | 50percent | 100percent | keepfirst | keepfirstlast |
> |---------------------|------------|----------:|-----------:|----------:|--------------:|
> |          gpt-4-0314 |      94.51 |     96.46 |      96.45 |     96.37 |         97.04 |
> |  gpt-3.5-turbo-0301 |      78.57 |     77.29 |      72.63 |     80.14 |         90.22 |
> |    text-davinci-003 |      80.13 |     72.11 |      37.35 |     78.75 |         88.95 |
> |      Llama-2-70b-hf |       71.0 |     44.93 |         <0 |     46.19 |          74.9 |
> | Llama-2-70b-chat-hf |      58.97 |     19.56 |         <0 |        <0 |         42.93 |
> |      Llama-2-13b-hf |      47.62 |     12.71 |         <0 |      0.99 |         43.59 |
> | Llama-2-13b-chat-hf |      23.35 |        <0 |         <0 |        <0 |         25.93 |
> |       Llama-2-7b-hf |      22.77 |        <0 |         <0 |        <0 |          9.88 |
> |  Llama-2-7b-chat-hf |       7.11 |        <0 |         <0 |        <0 |          2.39 |
> |          falcon-40b |      36.18 |        <0 |         <0 |        <0 |         12.71 |
> | falcon-40b-instruct |      30.56 |        <0 |         <0 |        <0 |         17.53 |
> |             mpt-30b |      46.96 |     19.08 |         <0 |     13.02 |         52.23 |
> |    mpt-30b-instruct |      28.46 |      2.49 |         <0 |      4.92 |         33.33 |
> |                 ul2 |      33.03 |        <0 |         <0 |        <0 |          5.22 |
> |            flan-ul2 |      24.21 |      7.13 |         <0 |        <0 |          1.72 |
> |         flan-t5-xxl |      38.68 |      12.4 |         <0 |        <0 |          2.22 |
> |            byt5-xxl |         <0 |        <0 |         <0 |        <0 |            <0 |
>
>
> **Scrambled QA on RealtimeQA (accuracy)**
> |                     |  original | 20percent | 50percent | 100percent | substituted | keepfirst | keepfirstlast |
> |---------------------|-----------|----------:|----------:|-----------:|------------:|----------:|--------------:|
> |          gpt-4-0314 |     98.18 |     97.26 |     96.05 |      93.01 |       62.92 |     93.92 |         95.74 |
> |  gpt-3.5-turbo-0301 |     97.26 |     96.66 |     91.19 |      84.50 |       55.62 |     86.93 |         91.49 |
> |    text-davinci-003 |     94.22 |     92.10 |     87.84 |      78.72 |       55.62 |     82.98 |         91.49 |
> |      Llama-2-70b-hf |     88.75 |     86.32 |     82.98 |      65.65 |       59.88 |     78.42 |         82.67 |
> | Llama-2-70b-chat-hf |     96.05 |     92.71 |     85.71 |      64.74 |       55.62 |     75.68 |         84.19 |
> |      Llama-2-13b-hf |     66.87 |     62.92 |     57.75 |      46.81 |       46.20 |     55.02 |         55.62 |
> | Llama-2-13b-chat-hf |     85.41 |     75.99 |     66.87 |      48.02 |       45.90 |     55.32 |         61.40 |
> |       Llama-2-7b-hf |     46.81 |     44.38 |     41.64 |      40.73 |       38.30 |     40.12 |         44.07 |
> |  Llama-2-7b-chat-hf |     82.67 |     77.51 |     65.05 |      46.20 |       46.20 |     50.76 |         60.79 |
> |          falcon-40b |     49.85 |     47.42 |     41.03 |      29.79 |       28.88 |     31.00 |         34.65 |
> | falcon-40b-instruct |     57.14 |     56.23 |     48.94 |      35.56 |       33.13 |     39.21 |         41.03 |
> |             mpt-30b |     62.31 |     56.53 |     50.15 |      41.64 |       39.51 |     47.42 |         50.15 |
> |    mpt-30b-instruct |     64.13 |     63.83 |     58.97 |      48.33 |       42.86 |     56.23 |         62.92 |
> |                 ul2 |     87.23 |     84.19 |     75.68 |      48.94 |       38.60 |     62.92 |         70.52 |
> |         flan-t5-xxl |     97.26 |     94.53 |     88.15 |      50.76 |       38.30 |     65.35 |         79.64 |
> |            byt5-xxl |     72.04 |     56.84 |     46.81 |      27.36 |       24.62 |     34.35 |         42.55 |
>
>
> **Scrambled QA on RealtimeQA (relative accuracy gain)**
> |                     |  original | 20percent | 50percent | 100percent | substituted | keepfirst | keepfirstlast |
> |---------------------|-----------|----------:|----------:|-----------:|------------:|----------:|--------------:|
> |          gpt-4-0314 |       1.0 |      0.97 |      0.94 |       0.85 |         0.0 |      0.88 |          0.93 |
> |  gpt-3.5-turbo-0301 |       1.0 |      0.99 |      0.85 |       0.69 |         0.0 |      0.75 |          0.86 |
> |    text-davinci-003 |       1.0 |      0.94 |      0.83 |       0.60 |         0.0 |      0.71 |          0.93 |
> |      Llama-2-70b-hf |       1.0 |      0.92 |      0.80 |       0.20 |         0.0 |      0.64 |          0.79 |
> | Llama-2-70b-chat-hf |       1.0 |      0.92 |      0.74 |       0.23 |         0.0 |      0.50 |          0.71 |
> |      Llama-2-13b-hf |       1.0 |      0.81 |      0.56 |       0.03 |         0.0 |      0.43 |          0.46 |
> | Llama-2-13b-chat-hf |       1.0 |      0.76 |      0.53 |       0.05 |         0.0 |      0.24 |          0.39 |
> |       Llama-2-7b-hf |       1.0 |      0.71 |      0.39 |       0.29 |         0.0 |      0.21 |          0.68 |
> |  Llama-2-7b-chat-hf |       1.0 |      0.86 |      0.52 |       0.00 |         0.0 |      0.13 |          0.40 |
> |          falcon-40b |       1.0 |      0.88 |      0.58 |       0.04 |         0.0 |      0.10 |          0.28 |
> | falcon-40b-instruct |       1.0 |      0.96 |      0.66 |       0.10 |         0.0 |      0.25 |          0.33 |
> |             mpt-30b |       1.0 |      0.75 |      0.47 |       0.09 |         0.0 |      0.35 |          0.47 |
> |    mpt-30b-instruct |       1.0 |      0.99 |      0.76 |       0.26 |         0.0 |      0.63 |          0.94 |
> |                 ul2 |       1.0 |      0.94 |      0.76 |       0.21 |         0.0 |      0.50 |          0.66 |
> |         flan-t5-xxl |       1.0 |      0.95 |      0.85 |       0.21 |         0.0 |      0.46 |          0.70 |
> |            byt5-xxl |       1.0 |      0.68 |      0.47 |       0.06 |         0.0 |      0.21 |          0.38 |
>
>
> **Scrambled QA on DREAM (accuracy)**
> |                     |  original | 20percent | 50percent | 100percent | substituted | keepfirst | keepfirstlast |
> |---------------------|-----------|----------:|----------:|-----------:|------------:|----------:|--------------:|
> |          gpt-4-0314 |     96.68 |     96.29 |     94.73 |      88.48 |       45.70 |     93.95 |         96.48 |
> |  gpt-3.5-turbo-0301 |     94.92 |     90.23 |     81.05 |      44.73 |       21.29 |     73.44 |         85.74 |
> |    text-davinci-003 |     93.36 |     91.02 |     81.84 |      54.69 |       45.12 |     79.30 |         88.48 |
> |      Llama-2-70b-hf |     92.97 |     90.62 |     83.40 |      57.23 |       50.20 |     74.02 |         85.16 |
> | Llama-2-70b-chat-hf |     91.02 |     86.13 |     76.95 |      48.44 |       45.70 |     59.77 |         74.41 |
> |      Llama-2-13b-hf |     70.51 |     65.62 |     54.69 |      40.82 |       38.28 |     42.38 |         52.93 |
> | Llama-2-13b-chat-hf |     80.08 |     74.80 |     62.89 |      42.58 |       41.02 |     48.05 |         58.59 |
> |       Llama-2-7b-hf |     60.74 |     55.66 |     48.83 |      38.48 |       38.67 |     42.97 |         46.29 |
> |  Llama-2-7b-chat-hf |     75.00 |     66.21 |     57.03 |      40.04 |       38.09 |     42.97 |         52.93 |
> |          falcon-40b |     64.06 |     59.77 |     53.52 |      44.34 |       42.77 |     48.63 |         54.69 |
> | falcon-40b-instruct |     73.05 |     65.62 |     57.23 |      42.97 |       43.16 |     48.63 |         57.03 |
> |             mpt-30b |     76.37 |     73.24 |     60.74 |      47.85 |       45.31 |     50.78 |         58.98 |
> |    mpt-30b-instruct |     86.13 |     78.52 |     69.53 |      45.12 |       42.77 |     55.47 |         63.09 |
> |                 ul2 |     44.14 |     41.02 |     33.20 |      20.31 |       17.77 |     22.27 |         28.71 |
> |            flan-ul2 |     94.53 |     91.02 |     82.23 |      53.71 |       49.22 |     65.23 |         76.37 |
> |         flan-t5-xxl |     96.68 |     94.34 |     86.72 |      54.49 |       53.52 |     70.70 |         84.77 |
>
>
> **Scrambled QA on DREAM (relative accuracy gain)**
> |                     |  original | 20percent | 50percent | 100percent | substituted | keepfirst | keepfirstlast |
> |---------------------|-----------|----------:|----------:|-----------:|------------:|----------:|--------------:|
> |          gpt-4-0314 |       1.0 |      0.99 |      0.96 |       0.84 |         0.0 |      0.95 |          1.00 |
> |  gpt-3.5-turbo-0301 |       1.0 |      0.94 |      0.81 |       0.32 |         0.0 |      0.71 |          0.88 |
> |    text-davinci-003 |       1.0 |      0.95 |      0.76 |       0.20 |         0.0 |      0.71 |          0.90 |
> |      Llama-2-70b-hf |       1.0 |      0.95 |      0.78 |       0.16 |         0.0 |      0.56 |          0.82 |
> | Llama-2-70b-chat-hf |       1.0 |      0.89 |      0.69 |       0.06 |         0.0 |      0.31 |          0.63 |
> |      Llama-2-13b-hf |       1.0 |      0.85 |      0.51 |       0.08 |         0.0 |      0.13 |          0.45 |
> | Llama-2-13b-chat-hf |       1.0 |      0.86 |      0.56 |       0.04 |         0.0 |      0.18 |          0.45 |
> |       Llama-2-7b-hf |       1.0 |      0.77 |      0.46 |      -0.01 |         0.0 |      0.19 |          0.35 |
> |  Llama-2-7b-chat-hf |       1.0 |      0.76 |      0.51 |       0.05 |         0.0 |      0.13 |          0.40 |
> |          falcon-40b |       1.0 |      0.80 |      0.50 |       0.07 |         0.0 |      0.28 |          0.56 |
> | falcon-40b-instruct |       1.0 |      0.75 |      0.47 |      -0.01 |         0.0 |      0.18 |          0.46 |
> |             mpt-30b |       1.0 |      0.90 |      0.50 |       0.08 |         0.0 |      0.18 |          0.44 |
> |    mpt-30b-instruct |       1.0 |      0.82 |      0.62 |       0.05 |         0.0 |      0.29 |          0.47 |
> |                 ul2 |       1.0 |      0.88 |      0.59 |       0.10 |         0.0 |      0.17 |          0.41 |
> |            flan-ul2 |       1.0 |      0.92 |      0.73 |       0.10 |         0.0 |      0.35 |          0.60 |
> |         flan-t5-xxl |       1.0 |      0.95 |      0.77 |       0.02 |         0.0 |      0.40 |          0.72 |
>
> > Typos, Grammar, Style: The writing of the paper should considerably be improved and grammatical errors should be corrected.
>
> Thank you so much for your careful inspection! We will definitely correct them in modified paper.

---

### Official Review · Reviewer_NrTg · 2023-08-11

**Soundness:** 4

**Excitement:**

4: Strong: This paper deepens the understanding of some phenomenon or lowers the barriers to an existing research direction.

**Paper Topic And Main Contributions:**

This paper focuses on the ability of LLMs to handle character-level scrambled text. The researchers created a benchmark and conducted experiments to test this, finding that GPT-4 excels at understanding scrambled text, even when the letters of all words are scrambled. Other LLMs perform well too, but GPT-4 stands out for its remarkable performance in processing inputs with unnatural errors.

**Questions For The Authors:**

Why GPT-4 performs better than text-davinci-003 and GPT-3,5-turbo?

**Reasons To Accept:**

1. This paper proposes a novel task
2. This paper proposes a benchmark for the corresponding task
4. Experiments on three models seem interesting. Never image that tokenizer-based models are immune to the character permutation.

**Reasons To Reject:**

1.  Did not test the performance of character-based/tokenizer-free models such as T5.
2. Only test on models that are not public. It would be interesting to test the benchmark on public models such as LLaMa and see the distribution, tokenization results, and other statistics for further analysis.
3. No further analysis.

**Reproducibility:**

5: Could easily reproduce the results.

**Reviewer Confidence:**

4: Quite sure. I tried to check the important points carefully. It's unlikely, though conceivable, that I missed something that should affect my ratings.

---

> ### Author Rebuttal · Authors · 2023-08-29
>
> Dear Reviewer NrTg:
>
> Thank you very much for your review.
>
> > 1. Did not test the performance of character-based/tokenizer-free models such as T5.
>
> We add T5 series models, including byt5-xxl (tokenizer-free). The original T5 is not included in results, since it is completely not capable in this setting.
>
> > 2. Only test on models that are not public. It would be interesting to test the benchmark on public models such as LLaMa and see the distribution, tokenization results, and other statistics for further analysis.
> > 3. No further analysis.
>
> Following the reviewers’ comments, we conducted experiments using various open-source LLMs, including (llama-2, falcon, mpt, ul2, flan-ul2, flan-t5, and byt5). The model covers diverse model architectures (decoder only and encoder-decoder), model sizes (from 7b to 70b), and training objectives (e.g., with or without further finetuning) and tokenizers (e.g., tokenizer-free: byt5-xxl). The results are shown below:
>
> **Scrambled Sentence Recovery on RealtimeQA (edit distance)**
> |                     |  20percent | 50percent | 100percent | keepfirst | keepfirstlast |
> |---------------------|------------|----------:|-----------:|----------:|--------------:|
> |          gpt-4-0314 |       1.49 |      2.40 |       4.79 |      3.79 |          2.24 |
> |  gpt-3.5-turbo-0301 |       5.80 |     15.38 |      36.87 |     20.76 |          7.40 |
> |    text-davinci-003 |       5.38 |     18.89 |      84.39 |     22.21 |          8.36 |
> |      Llama-2-70b-hf |       7.85 |     37.30 |     169.85 |     56.25 |         18.99 |
> | Llama-2-70b-chat-hf |      11.11 |     54.49 |     196.44 |    106.31 |         43.18 |
> |      Llama-2-13b-hf |      14.18 |     59.13 |     192.67 |    103.50 |         42.67 |
> | Llama-2-13b-chat-hf |      20.75 |     73.04 |     201.50 |    119.52 |         56.03 |
> |       Llama-2-7b-hf |      20.90 |     79.83 |     194.48 |    135.50 |         68.18 |
> |  Llama-2-7b-chat-hf |      25.14 |     83.13 |     196.88 |    132.44 |         73.84 |
> |          falcon-40b |      17.27 |     69.86 |     185.26 |    119.00 |         66.03 |
> | falcon-40b-instruct |      18.79 |     74.26 |     192.86 |    130.82 |         62.39 |
> |             mpt-30b |      14.36 |     54.81 |     181.43 |     90.92 |         36.14 |
> |    mpt-30b-instruct |      19.36 |     66.05 |     170.88 |     99.38 |         50.44 |
> |                 ul2 |      18.12 |     68.79 |     153.53 |    109.47 |         71.70 |
> |            flan-ul2 |      20.51 |     62.91 |     137.15 |    105.67 |         74.35 |
> |         flan-t5-xxl |      16.60 |     59.34 |     136.41 |    105.44 |         73.97 |
> |            byt5-xxl |      79.27 |    104.28 |     148.44 |    123.97 |        102.50 |
>
>
> **Scrambled Sentence Recovery on RealtimeQA(recovery rate)**
> |                     |  20percent | 50percent | 100percent | keepfirst | keepfirstlast |
> |---------------------|------------|----------:|-----------:|----------:|--------------:|
> |          gpt-4-0314 |      94.51 |     96.46 |      96.45 |     96.37 |         97.04 |
> |  gpt-3.5-turbo-0301 |      78.57 |     77.29 |      72.63 |     80.14 |         90.22 |
> |    text-davinci-003 |      80.13 |     72.11 |      37.35 |     78.75 |         88.95 |
> |      Llama-2-70b-hf |       71.0 |     44.93 |         <0 |     46.19 |          74.9 |
> | Llama-2-70b-chat-hf |      58.97 |     19.56 |         <0 |        <0 |         42.93 |
> |      Llama-2-13b-hf |      47.62 |     12.71 |         <0 |      0.99 |         43.59 |
> | Llama-2-13b-chat-hf |      23.35 |        <0 |         <0 |        <0 |         25.93 |
> |       Llama-2-7b-hf |      22.77 |        <0 |         <0 |        <0 |          9.88 |
> |  Llama-2-7b-chat-hf |       7.11 |        <0 |         <0 |        <0 |          2.39 |
> |          falcon-40b |      36.18 |        <0 |         <0 |        <0 |         12.71 |
> | falcon-40b-instruct |      30.56 |        <0 |         <0 |        <0 |         17.53 |
> |             mpt-30b |      46.96 |     19.08 |         <0 |     13.02 |         52.23 |
> |    mpt-30b-instruct |      28.46 |      2.49 |         <0 |      4.92 |         33.33 |
> |                 ul2 |      33.03 |        <0 |         <0 |        <0 |          5.22 |
> |            flan-ul2 |      24.21 |      7.13 |         <0 |        <0 |          1.72 |
> |         flan-t5-xxl |      38.68 |      12.4 |         <0 |        <0 |          2.22 |
> |            byt5-xxl |         <0 |        <0 |         <0 |        <0 |            <0 |
>
>
> **Scrambled QA on RealtimeQA (accuracy)**
> |                     |  original | 20percent | 50percent | 100percent | substituted | keepfirst | keepfirstlast |
> |---------------------|-----------|----------:|----------:|-----------:|------------:|----------:|--------------:|
> |          gpt-4-0314 |     98.18 |     97.26 |     96.05 |      93.01 |       62.92 |     93.92 |         95.74 |
> |  gpt-3.5-turbo-0301 |     97.26 |     96.66 |     91.19 |      84.50 |       55.62 |     86.93 |         91.49 |
> |    text-davinci-003 |     94.22 |     92.10 |     87.84 |      78.72 |       55.62 |     82.98 |         91.49 |
> |      Llama-2-70b-hf |     88.75 |     86.32 |     82.98 |      65.65 |       59.88 |     78.42 |         82.67 |
> | Llama-2-70b-chat-hf |     96.05 |     92.71 |     85.71 |      64.74 |       55.62 |     75.68 |         84.19 |
> |      Llama-2-13b-hf |     66.87 |     62.92 |     57.75 |      46.81 |       46.20 |     55.02 |         55.62 |
> | Llama-2-13b-chat-hf |     85.41 |     75.99 |     66.87 |      48.02 |       45.90 |     55.32 |         61.40 |
> |       Llama-2-7b-hf |     46.81 |     44.38 |     41.64 |      40.73 |       38.30 |     40.12 |         44.07 |
> |  Llama-2-7b-chat-hf |     82.67 |     77.51 |     65.05 |      46.20 |       46.20 |     50.76 |         60.79 |
> |          falcon-40b |     49.85 |     47.42 |     41.03 |      29.79 |       28.88 |     31.00 |         34.65 |
> | falcon-40b-instruct |     57.14 |     56.23 |     48.94 |      35.56 |       33.13 |     39.21 |         41.03 |
> |             mpt-30b |     62.31 |     56.53 |     50.15 |      41.64 |       39.51 |     47.42 |         50.15 |
> |    mpt-30b-instruct |     64.13 |     63.83 |     58.97 |      48.33 |       42.86 |     56.23 |         62.92 |
> |                 ul2 |     87.23 |     84.19 |     75.68 |      48.94 |       38.60 |     62.92 |         70.52 |
> |         flan-t5-xxl |     97.26 |     94.53 |     88.15 |      50.76 |       38.30 |     65.35 |         79.64 |
> |            byt5-xxl |     72.04 |     56.84 |     46.81 |      27.36 |       24.62 |     34.35 |         42.55 |
>
>
> **Scrambled QA on RealtimeQA (relative accuracy gain)**
> |                     |  original | 20percent | 50percent | 100percent | substituted | keepfirst | keepfirstlast |
> |---------------------|-----------|----------:|----------:|-----------:|------------:|----------:|--------------:|
> |          gpt-4-0314 |       1.0 |      0.97 |      0.94 |       0.85 |         0.0 |      0.88 |          0.93 |
> |  gpt-3.5-turbo-0301 |       1.0 |      0.99 |      0.85 |       0.69 |         0.0 |      0.75 |          0.86 |
> |    text-davinci-003 |       1.0 |      0.94 |      0.83 |       0.60 |         0.0 |      0.71 |          0.93 |
> |      Llama-2-70b-hf |       1.0 |      0.92 |      0.80 |       0.20 |         0.0 |      0.64 |          0.79 |
> | Llama-2-70b-chat-hf |       1.0 |      0.92 |      0.74 |       0.23 |         0.0 |      0.50 |          0.71 |
> |      Llama-2-13b-hf |       1.0 |      0.81 |      0.56 |       0.03 |         0.0 |      0.43 |          0.46 |
> | Llama-2-13b-chat-hf |       1.0 |      0.76 |      0.53 |       0.05 |         0.0 |      0.24 |          0.39 |
> |       Llama-2-7b-hf |       1.0 |      0.71 |      0.39 |       0.29 |         0.0 |      0.21 |          0.68 |
> |  Llama-2-7b-chat-hf |       1.0 |      0.86 |      0.52 |       0.00 |         0.0 |      0.13 |          0.40 |
> |          falcon-40b |       1.0 |      0.88 |      0.58 |       0.04 |         0.0 |      0.10 |          0.28 |
> | falcon-40b-instruct |       1.0 |      0.96 |      0.66 |       0.10 |         0.0 |      0.25 |          0.33 |
> |             mpt-30b |       1.0 |      0.75 |      0.47 |       0.09 |         0.0 |      0.35 |          0.47 |
> |    mpt-30b-instruct |       1.0 |      0.99 |      0.76 |       0.26 |         0.0 |      0.63 |          0.94 |
> |                 ul2 |       1.0 |      0.94 |      0.76 |       0.21 |         0.0 |      0.50 |          0.66 |
> |         flan-t5-xxl |       1.0 |      0.95 |      0.85 |       0.21 |         0.0 |      0.46 |          0.70 |
> |            byt5-xxl |       1.0 |      0.68 |      0.47 |       0.06 |         0.0 |      0.21 |          0.38 |
>
>
> **Scrambled QA on DREAM (accuracy)**
> |                     |  original | 20percent | 50percent | 100percent | substituted | keepfirst | keepfirstlast |
> |---------------------|-----------|----------:|----------:|-----------:|------------:|----------:|--------------:|
> |          gpt-4-0314 |     96.68 |     96.29 |     94.73 |      88.48 |       45.70 |     93.95 |         96.48 |
> |  gpt-3.5-turbo-0301 |     94.92 |     90.23 |     81.05 |      44.73 |       21.29 |     73.44 |         85.74 |
> |    text-davinci-003 |     93.36 |     91.02 |     81.84 |      54.69 |       45.12 |     79.30 |         88.48 |
> |      Llama-2-70b-hf |     92.97 |     90.62 |     83.40 |      57.23 |       50.20 |     74.02 |         85.16 |
> | Llama-2-70b-chat-hf |     91.02 |     86.13 |     76.95 |      48.44 |       45.70 |     59.77 |         74.41 |
> |      Llama-2-13b-hf |     70.51 |     65.62 |     54.69 |      40.82 |       38.28 |     42.38 |         52.93 |
> | Llama-2-13b-chat-hf |     80.08 |     74.80 |     62.89 |      42.58 |       41.02 |     48.05 |         58.59 |
> |       Llama-2-7b-hf |     60.74 |     55.66 |     48.83 |      38.48 |       38.67 |     42.97 |         46.29 |
> |  Llama-2-7b-chat-hf |     75.00 |     66.21 |     57.03 |      40.04 |       38.09 |     42.97 |         52.93 |
> |          falcon-40b |     64.06 |     59.77 |     53.52 |      44.34 |       42.77 |     48.63 |         54.69 |
> | falcon-40b-instruct |     73.05 |     65.62 |     57.23 |      42.97 |       43.16 |     48.63 |         57.03 |
> |             mpt-30b |     76.37 |     73.24 |     60.74 |      47.85 |       45.31 |     50.78 |         58.98 |
> |    mpt-30b-instruct |     86.13 |     78.52 |     69.53 |      45.12 |       42.77 |     55.47 |         63.09 |
> |                 ul2 |     44.14 |     41.02 |     33.20 |      20.31 |       17.77 |     22.27 |         28.71 |
> |            flan-ul2 |     94.53 |     91.02 |     82.23 |      53.71 |       49.22 |     65.23 |         76.37 |
> |         flan-t5-xxl |     96.68 |     94.34 |     86.72 |      54.49 |       53.52 |     70.70 |         84.77 |
>
>
> **Scrambled QA on DREAM (relative accuracy gain)**
> |                     |  original | 20percent | 50percent | 100percent | substituted | keepfirst | keepfirstlast |
> |---------------------|-----------|----------:|----------:|-----------:|------------:|----------:|--------------:|
> |          gpt-4-0314 |       1.0 |      0.99 |      0.96 |       0.84 |         0.0 |      0.95 |          1.00 |
> |  gpt-3.5-turbo-0301 |       1.0 |      0.94 |      0.81 |       0.32 |         0.0 |      0.71 |          0.88 |
> |    text-davinci-003 |       1.0 |      0.95 |      0.76 |       0.20 |         0.0 |      0.71 |          0.90 |
> |      Llama-2-70b-hf |       1.0 |      0.95 |      0.78 |       0.16 |         0.0 |      0.56 |          0.82 |
> | Llama-2-70b-chat-hf |       1.0 |      0.89 |      0.69 |       0.06 |         0.0 |      0.31 |          0.63 |
> |      Llama-2-13b-hf |       1.0 |      0.85 |      0.51 |       0.08 |         0.0 |      0.13 |          0.45 |
> | Llama-2-13b-chat-hf |       1.0 |      0.86 |      0.56 |       0.04 |         0.0 |      0.18 |          0.45 |
> |       Llama-2-7b-hf |       1.0 |      0.77 |      0.46 |      -0.01 |         0.0 |      0.19 |          0.35 |
> |  Llama-2-7b-chat-hf |       1.0 |      0.76 |      0.51 |       0.05 |         0.0 |      0.13 |          0.40 |
> |          falcon-40b |       1.0 |      0.80 |      0.50 |       0.07 |         0.0 |      0.28 |          0.56 |
> | falcon-40b-instruct |       1.0 |      0.75 |      0.47 |      -0.01 |         0.0 |      0.18 |          0.46 |
> |             mpt-30b |       1.0 |      0.90 |      0.50 |       0.08 |         0.0 |      0.18 |          0.44 |
> |    mpt-30b-instruct |       1.0 |      0.82 |      0.62 |       0.05 |         0.0 |      0.29 |          0.47 |
> |                 ul2 |       1.0 |      0.88 |      0.59 |       0.10 |         0.0 |      0.17 |          0.41 |
> |            flan-ul2 |       1.0 |      0.92 |      0.73 |       0.10 |         0.0 |      0.35 |          0.60 |
> |         flan-t5-xxl |       1.0 |      0.95 |      0.77 |       0.02 |         0.0 |      0.40 |          0.72 |
>
> For the further analysis, we intuitively thought that LLMs handle scrambled text by mapping the scrambled tokens to original ones. To investigate this, we analyze the hidden state of Llama-2 when inputting scrambled sentences. We calculate the cosine similarity of the hidden state (excluding embedding layers and last two layers, since they are strongly related to input and output, respectively) of each pair of scrambled sentences and original sentences. Then, we average the cosine similarity and normalize it from 0 (sentences with randomly substituted letters) to 1 (original sentences). The results are shown in the table below (it is difficult to show layer-wise comparison, since not figure allowed). For models of all three sizes, the similarity decreases through scrambling severity increases. And the similarity is not related to the performance on tasks. We also sort the samples by recovery rate and separate the samples into two groups based on the correctness on QA tasks and compare their similarities. However, no patterns can be found. So, we assume that LLMs handle scrambled sentences not by mapping them to original ones.
>
> |     | 10%  | 20%  | 30%  | 40%  | 50%  | 60%  | 70%  | 80%  | 90%  | 100% | kf   | kfl  |
> |-----|------|------|------|------|------|------|------|------|------|------|------|------|
> | 70b | 0.91 | 0.81 | 0.72 | 0.63 | 0.54 | 0.46 | 0.37 | 0.29 | 0.23 | 0.15 | 0.27 | 0.41 |
> | 13b | 0.92 | 0.81 |  0.7 | 0.59 | 0.49 | 0.39 |  0.3 | 0.22 | 0.15 | 0.09 | 0.18 | 0.32 |
> | 7b  | 0.97 | 0.92 | 0.86 |  0.8 | 0.73 | 0.67 |  0.6 | 0.53 | 0.46 | 0.39 | 0.49 |  0.6 |
>
> > Why GPT-4 performs better than text-davinci-003 and GPT-3,5-turbo?
>
> Because it’s difficult to access the models and there is no detailed description of models, it is difficult to draw conclusions about the reason why GPT-4 performs better. Since GPT-4 can do much better on these tasks, we assume this might be an emergent ability related to parameter size and the way of pre-training (e.g., one of possible reasons: enough contents with typos existing in training data).

---

### Meta-Review · Area_Chair_yfNU · 2023-09-25

**Recommendation:** 4

**Metareview:**

This paper explores the ability of large language models (LLMs) to handle character-level scrambled text. The authors created a benchmark and conducted experiments to test this. They found that GPT-4 excels at understanding scrambled text, even when the letters of words are scrambled. Other LLMs also perform well, but GPT-4 stands out for its remarkable performance in processing inputs with unnatural errors. The authors evaluated the performance of the GPT-4 method for two tasks: scrambled sentence recovery and scrambled question answering. They also propose two new datasets.  This work is interesting and well conducted.

Pros:
This paper addresses an interesting research question and proposes a benchmark for the corresponding task.
The paper is well-grounded in current literature, citing recent approaches and advances in the field.
The methodology seems simple but well thought-out. The  investigation could contribute to the probing of the abilities of LLMs.
The authors extended  the evaluation models and datasets during the rebuttal.
Most of the cons covered in the rebuttal.
Typos, proofreading is highly recommended.

---

### Decision · Program_Chairs · 2023-10-07

**Decision:**

Accept-Main

**Comment:**

This paper explores the ability of large language models (LLMs) to handle character-level scrambled text. The authors created a benchmark and conducted experiments to test this. They found that GPT-4 excels at understanding scrambled text, even when the letters of words are scrambled. Other LLMs also perform well, but GPT-4 stands out for its remarkable performance in processing inputs with unnatural errors. The authors evaluated the performance of the GPT-4 method for two tasks: scrambled sentence recovery and scrambled question answering. They also propose two new datasets.  This work is interesting and well conducted.

Pros:
This paper addresses an interesting research question and proposes a benchmark for the corresponding task.
The paper is well-grounded in current literature, citing recent approaches and advances in the field.
The methodology seems simple but well thought-out. The  investigation could contribute to the probing of the abilities of LLMs.
The authors extended  the evaluation models and datasets during the rebuttal.
Most of the cons covered in the rebuttal.
Typos, proofreading is highly recommended.